

# Implications of potential future grand solar minimum for ozone layer and climate

Pavle Arsenovic[1], Eugene Rozanov[1,2], Julien Anet[3], Andrea Stenke[1], Thomas Peter[1]

[1]Institute for Atmospheric and Climate Science, ETH, Zürich, Switzerland

[2]Physikalisch-Meteorologisches Observatorium Davos/World Radiation Center, Davos, Switzerland

[3]Zürich University of Applied Sciences, Winterthur, Switzerland

*Correspondence to*: Pavle Arsenovic (pavle.arsenovic@env.ethz.ch)

**Abstract.** Continued anthropogenic greenhouse gas (GHG) emissions are expected to cause further global warming throughout the 21[st] century. Understanding potential interferences with natural forcings is thus of great interest. Here we investigate the

impact of a recently proposed 21[st] century grand solar minimum on atmospheric chemistry and climate using the SOCOL3-MPIOM chemistry-climate model with interactive ocean. We examine several model simulations for the period 2000 – 2199, following the greenhouse gas scenario RCP4.5, but with different solar forcings: the reference simulation is forced by perpetual repetition of solar cycle 23 until the year 2199, whereas the grand solar minimum simulations assume strong declines in solar activity of 3.5 and 6.5 W/m[2] with different durations. Decreased solar activity is found to yield up to a doubling of the GHG

induced stratospheric and mesospheric cooling. Under the grand solar minimum scenario tropospheric temperatures are also projected to decrease. On the global scale the reduced solar forcing compensates at most 15% of the expected greenhouse warming at the end of 21[st] and around 25% at the end of 22[nd] century. The regional effects are predicted to be stronger, in particular in northern high latitude winter. In the stratosphere, the reduced incoming ultraviolet radiation leads to less ozone production by up to 8%, which overcompensates the anticipated ozone increase due to reduced stratospheric temperatures and

an acceleration of the Brewer-Dobson circulation. This, in turn, leads to a delay in total ozone column recovery from anthropogenic chlorine-induced depletion, with a global ozone recovery to the pre-ozone hole values happening only upon completion of the grand solar minimum in the 22[nd] century or later.

## 1 Introduction

Global warming is one of the main current societal problems. The observed global warming since preindustrial period (1850 – 1900) until the end of 20[th] century (1986 – 2005) is estimated to be around 0.6 °C (IPCC, 2013). The mean global surface temperature is expected to continue to rise in the 21[st] century due to human activity and an associated increase of greenhouse



gas (GHG) concentrations. In its fifth assessment report (AR5) the Intergovernmental Panel on Climate Change (IPCC) examined four Representative Concentration Pathways (RCPs) of GHG concentration trajectories (IPCC, 2013). The projected warming (2081-2100 mean minus 1986-2005 mean) is 1±0.4 °C for RCP2.6, 1.8±0.5 °C for RCP4.5, 2.2±0.5 °C for RCP6.0, and 3.7±0.7 °C RCP8.5, given as multi-model mean ± standard deviation of the various IPCC models. In December 2015,

many countries agreed to make an effort to reduce their emissions of GHG into the atmosphere in order to keep the global surface temperature rise below 2 °C above pre-industrial levels. This agreement was adopted under United Nations Framework Convention on Climate Change (UNFCCC), and it is now known as The Paris climate agreement. RCP2.6 is the only GHG concentration scenario that limits the global mean surface temperature increase at 2 °C at the end of 21[st] century (van Vuuren et al., 2011b).

A second major anthropogenic influence on the atmosphere results from the release of ozone depleting substances (ODSs). Rowland and Molina (1974) warned against human-produced chemicals playing an important role in stratospheric ozone depletion, leading to a thinning of the ozone layer, thereby increasing the incidents of skin cancer and eye cataracts, but also affecting plants, crops and the ocean ecosystem (e.g., Hegglin et al., 2015). Observations confirmed the global ozone depletion,

and revealed that the maximum ozone depletion occurred in the springtime Antarctic stratosphere, a phenomenon commonly known as the "ozone hole". As a response to ozone depletion, the Montreal Protocol was established in 1987, which prohibited emissions of certain ODSs into the atmosphere. In their latest report on the ozone layer, the World Meteorological Organization (WMO) and United Nations Environmental Programme (UNEP) projected that the reduction of ODSs will lead to ozone increase in the 21[st] century, reaching pre-1980 levels in the second half of the century, with detailed recovery times depending

on latitude (WMO, 2014).

The projections of terrestrial climate by the IPCC and of the ozone layer by WMO/UNEP assume solar irradiance to remain basically constant with respect to both incoming integrated power (total solar irradiance, TSI) and spectral distribution of the power (spectral solar irradiance, SSI). However, the Sun is a variable star and its output varies over vast time scales. The

decadal-scale solar variability was discovered in the middle of the 19[th] century (Schwabe, 1852; Wolf, 1861). This cycle is today known as the 11-year solar cycle and is characterized by a change in TSI between maximum and minimum of approximately 1 W/m$^2$ at Earth distance (Fröhlich, 2006), corresponding to about 0.07% of the average TSI of 1366 W/m$^2$ (Gray et al., 2010). Solar activity on longer time scales can be reconstructed using cosmogenic radionuclides, whose atmospheric production rate is modulated by solar activity. The reconstructions reveal cycles in the order of hundreds of years,

called "grand solar minima" and "grand solar maxima". Several recent publications suggest a new grand solar minimum to occur in the 21[st] century (Abreu et al., 2010; Lockwood et al., 2011; Roth and Joos, 2013) and to last even until the end of 22[nd] century (Steinhilber and Beer, 2013). Such events might have a significant impact on climate and on the ozone layer. As an example, the Dalton minimum (1790 – 1830) is thought to have contributed to significant cooling in Europe (Brugnara et al., 2013; Luterbacher et al., 2004).  It was characterized by reduced solar irradiation (Hoyt and Schatten, 1998), estimated to




range between a moderate ~1 W/m$^2$ (Kopp, 2016) and as much as ~5 W/m$^2$ (Shapiro et al., 2011) below present values. Anet et al. (2013b) applied the forcing derived by Shapiro et al. (2011) to modulate the solar input in a climate model and found that, among other natural factors (e.g. volcanic activity), the computed cooling was to a large degree caused by low solar activity. A grand solar minimum, which was even more prolonged than the Dalton Minimum was the Maunder Minimum, the
period starting around 1645 and continuing to around 1715 when sunspots became exceedingly rare.

Energetic particle precipitation (EPP) is closely related to solar activity. Energetic particles have the ability to produce odd nitrogen and odd hydrogen species, NO$_x$ ([N] + [NO] + [NO$_2$]) and HO$_x$ ([H] + [OH] + [HO$_2$]), which are known to catalytically deplete ozone. Amongst all energetic particles, galactic cosmic rays (GCR) are the most energetic (1 MeV to $5 \cdot 10^{13}$ MeV;
Dorman, 2004), so that they penetrate deep into the atmosphere. Their influence is most important in the polar lower stratosphere and upper troposphere (Calisto et al., 2011; Jackman et al., 2016; Mironova et al., 2015). Their intensity is anticorrelated with the solar activity (Bazilevskaya et al., 2008). Conversely, low energy electrons (LEE) are stopped already in the upper atmosphere and produce NO$_x$ in thermosphere, above 80 km altitude. During polar night, NO$_x$ created by LEE is then transported downwards and affects mesospheric and stratospheric ozone (Rozanov et al., 2012). Therefore, inclusion of
these processes in chemistry-climate models is important for a realistic representation of ozone.

With respect to surface temperature, reductions in solar activity may lead to a partial compensation of the radiative forcing stemming from increased anthropogenic emissions of GHGs. A number of studies were conducted to estimate if a potential future grand solar minimum would slow or even cancel global warming. Mokhov et al. (2008) performed 21[st] century
simulations with different solar, volcanic and anthropogenic forcings. Their analysis of the response of global mean near-surface temperature to various solar scenarios showed that solar activity variations of up to 2 W/m$^2$ impose only small changes in the surface temperature. Meehl et al. (2013) used the climate model CESM1 WACCM to investigate whether a future Maunder-like minimum could stop global warming. They found that such a potential grand solar minimum in the middle of 21[st] century would slow and delay anthropogenic global warming, such that surface temperatures would be lower by several
tenths of a degree by the end of the grand minimum. However, their study focused on surface temperature, whereas chemical effects and stratospheric changes caused by a grand solar minimum were not investigated. Another modelling study was performed by Anet et al. (2013a) using the SOCOL3-MPIOM model. Their work also showed a reduction of surface temperatures of the same order of magnitude as shown by Meehl et al. (2013) and a delay of the ozone recovery back to the "pre-ozone hole" conditions. Ineson et al. (2015) used the HadGEM2-CC climate model to evaluate possible impacts of a
grand solar minimum on climate. They found that the reduction of solar irradiance of about 0.13% would globally cool the surface by around 0.1 K for the second half of 21[st] century. Maycock et al. (2015) used the same model and applied a decrease in total solar irradiance and UV over the second half of 21[st] century of 0.12% and 0.85% respectively, compared to present values. They found that the decrease in solar activity would reduce global annual near surface temperature by around 0.1 K



and cool the stratopause region by around 1.2 K. However, their climate model lacked interactive chemistry, hence change in ozone and the influence of the EPP were neglected.

The present work is a continuation and extension of the study of Anet et al. (2013a). Here we investigate the atmospheric

response to a potential grand solar minimum which starts around 2020, reaches full depth by about 2090, and lasts either until 2200 or recovers within the 22$^{nd}$ century.

## 2 Methods

We use the coupled chemistry-climate model SOCOL3-MPIOM (Stenke et al., 2013; Muthers et al., 2014), which consists of the atmospheric model coupled to the chemistry module and the ocean model. The atmospheric component is a general

circulation model ECHAM5.4, a spectral model based on primitive equations with temperature, vorticity, divergence, the surface pressure, humidity and cloud water as prognostic variables (Manzini et al., 2006; Roeckner, 2003; Roeckner et al., 2006). Here it was applied in a configuration with T31 spectral horizontal truncation (approximately 3.75° x 3.75° horizontal resolution) and 39 vertical levels from the ground to 0.01 hPa (~80 km). The chemistry module is MEZON (Egorova et al., 2003; Rozanov et al., 1999), which computes the tendencies of 41 gas species, taking into account 200 gas-phase, 16

heterogeneous and 35 photolytical reactions. The oceanic component is MPIOM, a primitive equation model with the hydrostatic and Boussinesq assumptions made. It includes a dynamic/thermodynamic sea-ice module and uses a curvilinear orthogonal grid which allows for various setups. In our study it was used with a nominal horizontal resolution of 3°, divided vertically into 40 levels from the ocean surface to the bottom (for more details see Muthers et al., 2014).

We simulated five different scenarios, each with two ensemble members, with the only difference between these experiments being the applied solar forcing: four experiments with grand solar minima of two different strengths and two different durations, plus a reference simulation (see Figure 1). The reference simulation (hereafter termed REF) is forced by a perpetual repetition of the solar cycle 23 until the year 2199. Two experiments assume a weak drop (termed WD or WDR) in the solar forcing with total solar irradiance (TSI) approximately 3.5 W/m$^2$ lower than in REF (0.25% reduction). The assumed solar

minimum either continues throughout the 22$^{nd}$ century (WD) or starts to recover (WDR) soon after reaching the minimum of -3.5 W/m$^2$ around the year 2087. Two further experiments assume a strong drop (termed SD or SDR) with TSI about 6.5 W/m$^2$ lower (0.48% reduction) than in REF, again either continuing throughout the 22$^{nd}$ century (SD) or recovering (SDR) soon after reaching the minimum (Figure 1). Since this is a continuation of the study of Anet et al. (2013a), we are using the same solar forcing as they did. It is calculated using the method developed by Shapiro et al. (2011) based on the solar modulation potential

($\Phi$). The applied spectral solar irradiance (SSI) is described in Anet et al. (2013a, Figure S3). The prolonged grand solar minimum scenarios (WD and SD) are based on the same $\Phi$: WD represents the upper envelope of the uncertainty range of the solar forcing reconstruction, while SD represents the mean of solar forcing. The same applies to WDR and SDR, but the $\Phi$





follows the recovery of grand solar minimum. We call the scenarios "weak" and "strong" for clear distinction, though it must be noted that both scenarios actually represent stronger irradiance reductions than those generally assumed previous studies (Ineson et al., 2015; Maycock et al., 2015; Meehl et al., 2013; Mokhov et al., 2008) and by the IPCC. As described by Meehl et al. (2013), previous estimates regarding the TSI decrease during the Maunder Minimum compared to present-day values

range from somewhere close to present 11-year solar minima, to reductions of 0.15% to 0.3% below present solar minima all the way to more than 0.4% below present solar minima derived by Shapiro et al. (2011) and applied here in the SD and SDR scenarios. The stronger reductions in TSI have been criticized as being too large (Feulner, 2011), but here we regard these estimates as an absolute lower bound in TSI. Judge et al. (2012) found the Shapiro et al. (2011) estimates to be within bounds set by current stellar data, however, likely have over-estimated quiet-Sun irradiance variations by about a factor of two, based

upon a re-analysis of sub-mm data from the James Clerk Maxwell telescope. This is the basis for the WD and WDR scenarios employed here. In agreement with Meehl et al. (2013) we emphasize that the caveat for the present study is that an actual future Maunder Minimum-type event could feature a smaller reduction of TSI and an even lower climate system response.

To extend the simulations to the 22$^{nd}$ century, we repeated the last solar cycle for WD and SD simulated by Anet et al. (2013a)

for the year 2090 (Figure 1) until the year 2199 as was suggested by Steinhilber and Beer (2013). For WDR and SDR we mirrored SSI values backwards from 2088 into the future. This way we constructed the recovery of solar activity to pre-grand solar minimum values. The parameterizations of galactic cosmic rays (GCR), solar energetic protons and low energy electrons (LEE) were introduced identically to Rozanov et al. (2012). Apart from solar irradiance, $\Phi$ is used to parametrize GCR (based on Usoskin et al., 2010) and also to develop the geomagnetic activity ($A_p$) index needed for the LEE parameterization. As

mentioned above, since "weak" and "strong" scenarios are developed from the same $\Phi$, the EPP forcing is identical in WD and SD and in WDR and SDR scenarios.

Tropospheric aerosols are adapted from NCAR Community Atmospheric Model (CAM3.5) simulations with a bulk aerosol model forced with CMIP4 sea surface temperatures and the 2000 – 2100 CMIP5 emissions. For the 22$^{nd}$ century simulations,

they are fixed at 2090 levels. Stratospheric aerosols are kept at background levels except for the following four randomly chosen volcanic eruptions in the 21$^{st}$ century: Fuego-like eruption in 2024, a smaller eruption in 2033, an Agung-like eruption in 2060 and again a smaller eruption in 2073 (Anet et al., 2013a; 2013b). For the 22$^{nd}$ century we assume four identical small eruptions (with a magnitude between the eruptions in 2033 and 2073) in the years 2115, 2137, 2166 and 2187. The concentrations of GHGs and ODSs follow the CMIP5 RCP4.5 scenario (Meinshausen et al., 2011; van Vuuren et al., 2011a),

while the quasi-biennial oscillation (QBO) wind fields are nudged (for more details on the experimental set-up see Anet et al., 2013a).





## 3 Results

In order to understand the influence of a future grand solar minimum on climate and ozone layer evolutions, we first investigate the future evolution for the individual solar forcing scenarios. Subsequently we calculate differences in various quantities between the applied solar scenarios and REF for the future (2090 – 2099) to elucidate the role of the solar forcing.

### 3.1 Brewer-Dobson Circulation (BDC)

The transformed Eulerian mean vertical residual velocity ($w^*$; Hardiman et al., 2010) can be used as a measure of intensity of the Brewer-Dobson circulation. Figure 2 shows annual mean $w^*$ slightly above tropopause (70 hPa) averaged over 20° N – 20° S, since its maximum is around 15° – 20° on both hemispheres (Eyring et al., 2010). To reduce variability, the curves are smoothed with Savitzky and Golay (1964) filter.

The BDC accelerates in all experiments. From 1960 throughout the 21st century, the increase is 2 – 3% per decade, which agrees with SPARC multi-model mean (Eyring et al., 2010) and study of Butchart et al. (2006). The intensification is most evident in the first half of 21st century, when the increase of GHG concentrations is highest (van Vuuren et al., 2011, Figure 9). The second half of 21st century and the 22nd century show a continued acceleration of the BDC, however with a reduced strength, but statistically significant (at 95% confidence level using Mann-Kendall significance test). The intensification is highest in REF simulation and lower in WD and SD scenarios. However, in WDR and SDR scenarios, after the recovery of solar activity, BDC quickly adjusts to match the REF scenario at the end of 22nd century.

### 3.2 NOx response

According to the applied RCP4.5 scenario, surface emissions of $NO_x$ are decreasing and concentrations of $N_2O$ are increasing
during the 21st century. Decreasing $NO_x$ emissions in the troposphere lead to lower $NO_x$ concentrations in REF by up to 80% by the end of 21st century throughout the northern and of up to 40% throughout the southern troposphere (Figure 3a). In contrast, increasing $N_2O$ concentrations lead to increasing $NO_x$ levels in the upper stratosphere due to $N_2O$ conversion to reactive nitrogen oxides through the reaction with $O(^1D)$ (Brasseur and Solomon, 2005). By the end of the 21st century stratospheric $NO_x$ is projected to increase by about 10%. The $NO_x$ decrease by 20% in the tropical upper troposphere most
likely comes from decreasing tropospheric $NO_x$ and therefore less $NO_x$ transport from the troposphere into the stratosphere.

Figure 3b shows the simulated future change in $NO_x$ volume mixing ratio changes for the SD scenario. The different solar forcing leaves $NO_x$ levels unchanged in the troposphere, where $NO_x$ is dominated by anthropogenic influence. In the stratosphere, however, the effect of the solar irradiance decrease is well visible. Reduced NO photolysis limits $NO_x$ removal
in the stratosphere via the reaction $N + NO \rightarrow N_2 + O$, leading to a more pronounced stratospheric $NO_x$ increase under SD




than under REF conditions. Furthermore, the GCR intensity is stronger during grand solar minima leading to enhanced $NO_x$ production in the lower polar stratosphere. This effect, together with faster transport to the polar regions via BDC, yield around 50% more $NO_x$ the in southern and 20% in the northern hemisphere. During the grand solar minimum, the precipitation of LEE is decreased, leading to 60% reduced production of $NO_x$ in the polar mesosphere.

At the end of the 21$^{st}$ century, the lower photolysis rates in SD relative to REF would yield around 10% more stratospheric $NO_x$ (Figure 3c). The WD scenario would only lead to a 5% more stratospheric $NO_x$, i.e. about half the effect of SD (Figure 3d). As the applied LEE forcing in the model is the same for WD and SD, there is a similar reduction of $NO_x$ of around 80% in the polar mesosphere.

**3.3 Temperature response**

In REF, anthropogenic forcings according to RCP4.5 lead to a warming of the troposphere and a cooling of the stratosphere and mesosphere in the future as indicated in Figure 4a. The tropospheric warming reaches a maximum of around 3 K in the tropical upper troposphere. Zubov et al. (2013) showed that tropospheric warming is mainly caused by the surface warming due to increase of down-welling infrared radiation by GHG, enhanced by latent heat release in the middle troposphere. The
temperature decrease in the stratosphere and mesosphere results from increased cooling rates of GHGs. The secondary maximum in the Antarctic lower stratosphere is explained by the ozone recovery following the limitation of ODSs emissions. Li et al. (2009) used the Goddard Earth Observing System chemistry climate model to evaluate temperature and ozone response to GHG increases and ODS declines. They found a warming of the troposphere in the second half of the 21$^{st}$ century of up to 4 K compared to the mean 1975 – 1984 values, accompanied by a cooling of stratosphere of up to 8 K. Our results for the same
period (not shown) agree very well with their study. In the SD scenario the warming of the whole troposphere continues into the 22$^{nd}$ century relative to the end of 21$^{st}$, with an additional warming peaking at around 1 K in tropics (not shown), while stratospheric temperatures do not further change during that century. The latter is expected, as the stratosphere has relatively short thermal relaxation time of less than a month (Newman and Rosenfield, 1997).

Figure 4b shows the temperature difference between future and present for the SD scenario. Relative to REF in Figure 4a, the temperature response pattern shows a reduced warming by up to 1 K in the troposphere, but a more intensive cooling in the stratosphere and mesosphere. The analysis of the zonal annual mean temperature presented by Maycock et al. (2015) showed the most intense cooling around the stratopause of up to 1.5 K for the 2050 – 2099 period. Our results for the same period suggest a similar temperature pattern (not shown). However, the magnitude is larger: the most pronounced cooling is located
above the stratopause and amounts to around 2 K in the WD and 3 K in the SD scenario. The difference in magnitude is related to a smaller decrease of the solar UV irradiance in their study.



The temperature anomaly in SD compared to REF is shown in Figure 4c. The temperature difference increases from the tropopause to the mesopause up to -7 K. Cooling in the troposphere of around 0.5 K is also found, but this can compensate less than 20% of the warming caused by anthropogenic GHG emissions. As expected, the WD scenario (Figure 4d) shows a similar difference pattern, but with a smaller magnitude.

The global mean surface temperature evolution is displayed in Figure 5. As shown by Anet et al. (2013a), the global mean surface temperature rises in the 21st century in all three scenarios (REF, WD, SD). The difference of global surface temperature between REF and SD, averaged over the last 20 years of the 21st century, is about 0.3 K (as also found by Anet et al. (2013a)). Should the grand solar minimum persist until the end of the 22nd century, the difference between REF and SD would increase to about 0.6 K (averaged over 2180 – 2199), which is about 25% of projected global warming of 2.3 K at the end of the 22nd century compared to the base period (1986 – 2005). The continued temperature increase after 2100 is supported by the thermal inertia of the ocean, as all forcings are kept constant in the 22nd century.

In case of a recovery of the minimum within the 22nd century, the difference of global surface temperature between REF and SDR, averaged over the last 20 years of the 22nd century, is computed to be only about 0.1K. This temperature response would compensate just ~4% of the anthropogenic temperature increase at the end of 22nd (2180 – 2199) century. In other words, an occurrence of the grand solar minimum in 21st century followed by its recovery would only slightly reduce global surface temperature.

20   For the RCP4.5 scenario, climate models also predict a warming about 2 K at the end of 21st century (2081 – 2100) compared to the 1986 – 2005 reference period (Figure 12.8, IPCC, 2013). As the ocean has a larger heat capacity and thermal inertia than the land surface and the atmosphere, the warming over land is more pronounced. The increase is most prominent near the poles of both northern (up to 5 K) and southern hemisphere (up to 3 K) (Figure 12.11, IPCC, 2013), a feature known as polar amplification (Serreze and Barry, 2011). Comparing the end of the 21st century (2090 – 2099) to its beginning (2000 – 2009), our model reproduces the polar amplification well: REF yields an increase of up to 4 K in North America and of up to 2 K in Antarctica (Figure 6a). The other continental regions warm up by around 2 K, while the sea surface temperature increases by 1 – 1.5 K. Recent studies (IPCC, 2013; Bakker et al., 2016) suggest that the Atlantic Meridional Overturning Circulation (AMOC) could weaken in the 21st century resulting in a temperature reduction in the North Atlantic. Our simulations reproduce this characteristic cooling of 1 K in the northern Atlantic that could be caused by a weakening of the AMOC in the 21st century.

30

A pronounced global warming would persist even if a strong irradiance drop, scenario SD, occurred in the near future, see Figure 6b. The model suggests a reduction of the warming in northern high latitudes, i.e. the damping of the polar amplification in the SD scenario. The surface temperature increase is also damped over continental Africa, Asia and North America, but amplified around Antarctica. The sea surface temperature, although still increasing, shows a smaller warming compared to





REF. A study by Menary and Scaife (2014) suggests that the low solar irradiance might cause a strengthening of the AMOC through stratosphere-troposphere coupling. Our results confirm the disappearance of the cooling in the northern Atlantic, likely caused by a recovering AMOC in the grand solar minimum (Muthers et al., 2016). Beyond the scope of our paper, this phenomenon needs to be investigated in more detail as it might have an impact on global, and especially on the European

climate (Jackson et al., 2015).

The surface temperature continues to rise in 22$^{nd}$ century (Figure 6c and 6d) in both the REF and SD scenarios. REF shows that the further increase is located mostly on the continents, but also in the Pacific, Kamchatka, Alaska and Greenland. The warming patterns in the 22$^{nd}$ century show similar locations of maxima as at the end of the 21$^{st}$, but with a smaller magnitude.

The warming during the 22$^{nd}$ century is less pronounced than in the previous century, especially in the SD case. The maximum warming of 1 K is located in the high latitudes.

The reduction of the annual mean surface temperatures due to the reduced solar activity at the end of 21$^{st}$ century is pronounced in the Arctic, and generally continental areas show lower surface temperatures, except for Australia and Europe, see Figure 7a

for the strong reduction (SD) scenario. The sea surface temperatures decrease by up to 0.5 K. The weaker WD scenario also shows a reduced warming, but to a smaller magnitude (Figure 7b). The cooling is most prominent in Russia and North America, amounting to around 1 K. The temperature decrease over sea is confined to the Indian Ocean and does not exceed 0.5 K. In both scenarios simulating a solar anomaly, a temperature increase of up to 1 K in SD and 0.75 K in WD, is predicted over the North Atlantic, likely due to a partial recovery of AMOC (Muthers et al., 2016). Considering the SD scenario, the boreal winter

analysis of surface temperature response (Figure 7c and 7d) shows the largest temperature reduction in winter over the Barents Sea and northern Asia. Similar cooling areas appear also in the WD case. The warming in the North Atlantic and in Greenland is present in the boreal winter projection as well as in the annual mean. Ineson et al. (2015) showed wintertime cooling in northern Eurasia and eastern North America with minima of -1.5 K in the 2050 – 2099 period. Our simulations show a similar, but more pronounced pattern for the same period (not shown), possibly due to our applied drop in UV forcing being stronger

than the one used by Ineson et al. (2015). Chiodo et al. (2016) reported significant cooling in their model simulations in boreal winter in continental Asia and the Bering Sea with peaks of -1.2 K for 2005 – 2065. This cooling is accompanied by a warming in North America and off the coast of Japan. Our results for WD suggest a slight warming at the east coast of North America and Europe, albeit not statistically significant (not shown).

### 3.4 Ozone response

Due to the Montreal Protocol ODSs' concentrations are projected to further decrease in future, which is expected to lead to a recovery of stratospheric ozone, mainly in polar lower stratosphere and globally in the upper stratosphere (Figure 8a). The decrease in concentrations of chlorine species strongly affects polar lower stratospheric ozone (exceeding +30%), mainly due




to a slowing of heterogeneous chlorine chemistry in the polar winter stratosphere, which is also responsible for the Antarctic "ozone hole" (Solomon et al., 1986). The increase in the upper stratosphere of 15 – 20% is a result of reduced intensity of the catalytic ozone destruction by reactive chlorine species (Molina and Rowland, 1974). In particular in the tropical stratosphere, the increase in ozone is also due to the GHG induced cooling, which slows the catalytic ozone destruction cycles as well as

the reaction $O + O_3 \rightarrow 2\,O_2$. In the mesosphere the reaction $O + O_2 + M \rightarrow O_3 + M$ also becomes important as its reaction rate coefficient increases with cooling (Jonsson et al., 2004), leading to ozone increase of around 5%. Conversely, the future decline of $NO_x$ surface emissions will result in less tropospheric ozone with a maximum in the northern hemisphere of up to 20%.

Besides chemical processes, which depend on ODS concentrations and on temperature, the circulation changes expected to
result from GHG-induced radiative changes, are also important for ozone. The acceleration of the BDC causes faster transport of ozone from the tropics to high latitudes causing ozone decrease in the tropical lower stratosphere exceeding 10% around 100 hPa (Figure 8a). The further acceleration of the BDC during the 22$^{nd}$ century leads to a further reduction of tropical ozone by 5% (years 2190 – 2199 relative to 2090 – 2099, not shown) and an increase in polar regions of 5%.

The strong solar minimum scenario SD shows a similar ozone pattern (Figure 8b). The increase of ozone in the lower polar stratosphere is the same as in REF as the impact by ODSs does not seem to depend much on the solar activity. However, in the upper stratosphere the ozone increase is smaller than in the reference case, as its production is supressed by low level of solar UV. Regardless, decreasing ODSs concentrations dominate over a decrease of the solar activity, therefore stratospheric ozone mixing ratios increase. The ozone decrease in the troposphere and in the tropical stratosphere is very similar as in REF,
as it is a result of anthropogenic activities. The most pronounced future differences between ozone in the SD scenario and REF occur in the mesosphere. Reduced photolysis of water vapour results in future decreases of $HO_x$ of 40% in the mesosphere (not shown), which contribute to the ozone increase at these altitudes. Together with the $NO_x$ decline due to the LEE weakening in the grand solar minimum and the GHG-induced cooling, it leads to an increase of ozone in the mesosphere of up to 35%.

The comparison between SD and REF at the end of 21$^{st}$ century is depicted in Figure 8c. Due to the weaker solar UV irradiance in the grand solar minimum, stratospheric ozone is reduced of up to 8% in nearly the entire stratosphere in SD compared to REF scenario. Less mesospheric $NO_x$ and $HO_x$ and colder temperature in the grand solar minimum impacts ozone at these altitudes, leading to an increase of up to 30%. The results are similar in case of WD, but changes are smaller by approximately a factor of 2 (Figure 8d).


Figure 9a shows the future increase in the annual mean total ozone column (TOC) over the middle to high latitudes in REF, which is attributed to reduced emissions of ODSs and an enhanced BDC in the warmer climate (Zubov et al., 2013). This future increase reaches 40 (60) Dobson units in the Northern (Southern) Hemispheres, which corresponds to about 10 – 20% of TOC increase. Acceleration of the BDC is expected to transport more ozone from the tropics to mid-latitudes fostering



extra-tropical ozone recovery, but delaying ozone recovery in the tropics (Austin and Wilson, 2006; Shepherd, 2008; Waugh, 2009). Because of this effect, future tropical ozone levels even show further decline at the end of 21$^{st}$ century compared to near present values. The slowing of photochemical ozone loss reactions caused by the cooling in the stratosphere (Barnett et al., 1975; Jonsson et al., 2004) contributes to a smaller degree to the overall TOC evolution (Zubov et al., 2013). Li et al. (2009)

5 showed that the BDC acceleration plays a crucial role in future ozone recovery and spatial distribution. They found recovery of extratropical ozone in year 2060 to 1975-1984 levels, but the tropical TOC did not recover. A study performed by Shepherd (2008) also showed this so-called "super-recovery" of extra-tropical and "sub-recovery" of tropical ozone by the end of the 21$^{st}$ century with respect to 1960 values.

10 Figure 9b illustrates future TOC changes for the SD scenario. Reduction of the solar activity in the future changes the situation dramatically. Weaker solar UV reduces oxygen photolysis leading to lower ozone production rate and pronounced TOC depletion in the entire tropical area by around 15 – 20 DU ($\triangleq$ 5 – 6%). On the other hand, cooler ocean surface due to less solar activity (see Figure 7a) slightly reduces BDC relative to REF (see Figure 2). These two processes cancel about 30 – 50% of the TOC increase in the Northern hemisphere obtained for REF. Over the Southern hemisphere the TOC changes are

15 dominated by the reduction in ODSs (Zubov et al., 2013), therefore the implications of the potential solar minimum are not as dramatic.

The effect of a decrease of the solar activity on the TOC is illustrated in Figure 10a and 10b. Both of the grand solar minimum scenarios predict a reduction in TOC, which would be stronger in the SD than in the WD scenario. An ozone reduction of

20 around 10 DU in the tropics in SD and 5 DU in WD is mostly a result of reduced production, and to a lesser degree because of a very small difference in BDC between the experiments. The most affected areas are mid-latitudes with maximum around 20 DU in the SD case (up to 4%). Since the polar vortex prevents mixing of ozone-rich air with polar air, ozone-rich air accumulates in the mid-latitudes. We found that drop in solar activity deaccelerates polar vortices on both hemispheres (not shown) and due to the weaker polar vortex, more ozone is able to reach polar areas. Also, during the grand solar minimum,

25 less ozone is produced in the tropical lower stratosphere. These two factors lead to less accumulation of the ozone-rich air in the mid-latitudes, creating a TOC minimum.

The amount of UV radiation that reaches the surface depends on incoming UV as well as on ozone layer thickness. Although the solar UV input is reduced in grand solar minimum, we showed that the ozone layer is thinning in the tropical areas. The

30 increase in tropospheric O($^1$D) in the grand solar minimum (not shown) suggests that ozone photolysis by UV ($\lambda$<320 nm) is enhanced through reaction $O_3 + h\nu \rightarrow O_2 + O(^1D)$ (Brasseur and Solomon, 2005). The increase of UV radiation at ground level can have potential positive and negative effects on human health (Reichrath, 2006). UV radiation is important for the production of vitamin D and therefore for human health (Hart et al., 2011), but can also cause skin cancer (Armstrong and





Kricker, 2001). Furthermore, UV radiation was shown to be harmful to plants as well, damaging DNA, proteins, lipids and membranes (Hollosy, 2002).

A future grand solar minimum could delay the recovery of the ozone layer (Anet et al., 2013) by several years. In Figure 11
we show annual global mean total ozone column evolution until the end of the $22^{nd}$ century. The first decline in 1960 – 1990 period of total ozone is caused by the emission of ODSs before the Montreal Protocol coming into force. In the beginning of the $21^{st}$ century, with the Montreal Protocol being effective, total ozone is increasing in all three solar scenarios. In the second half of the century, after a substantial reduction of reactive halogen containing species, the dominant effect on ozone will be solar activity. In the reference scenario (REF), the total global ozone recovers to the 1960 – 1980 values and even exceeds
them. However, neither the weak nor the strong solar minimum scenario (WD and SD) show a recovery within the simulated period. If the grand solar minimum persists during the $22^{nd}$ century, as the stratospheric temperatures and solar UV irradiance stay unchanged, so does the global mean of TOC. However, since the BDC continues to accelerate, it will continue to redistribute ozone from the tropics to the polar regions, which lead to the absence of strong trends in the global mean value. When the grand minimum recovers, TOC recovers to REF values readily.

**4 Conclusions and Outlook**

In this paper we investigated the influence of a potential future grand solar minimum on atmospheric chemistry and climate. Such an event, should it occur with the extreme intensity assumed here, could partly counteract the anthropogenic climate change caused by ongoing emissions of greenhouse gases that follow RCP4.5 scenario, but would still be by far too weak to fully compensate it. Even if the grand solar minimum were fully developed by the year 2090 and then lasted until the end of
the $22^{nd}$ century, global mean surface temperatures would continue to rise. While the solar effect, when assuming the said strong drop in solar irradiance (the SD scenario), is predicted to compensate about 15% of GHG-induced warming by the year 2100, this fraction could increase to about 25% during the $22^{nd}$ century, suggesting that the Earth system is still equilibrating to the increased GHG concentrations (which stay approximately constant during the $22^{nd}$ century within RCP4.5). For the lowest GHG concentration scenario, RCP2.6, IPCC (2013) multi-model mean projects global warming at the end of $21^{st}$
century of 1±0.4 °C. Our results show that even in this case, the extreme drop in solar activity would only reduce the projected increase in surface temperature by around 20 – 50%. As expected, for the higher RCPs 6.0 and 8.5 the grand solar minimum would result in only very minor reduction of the warming. This leads us to conclusion that strong drop in solar forcing would help us reach the Paris agreement goal for RCP2.6 and increase the chance of reaching it for RCP4.5. Nevertheless, the multi-model mean of RCP4.5 (IPCC, 2013) would still be above 2 °C threshold.


Areas with the highest partial compensation of global warming are located in high northern latitudes especially in the winter period. Our results suggest that a grand solar minimum could lead to a recovery of AMOC, which might cancel the cooling in



North Atlantic in the 21$^{st}$ century (Muthers et al., 2016). More research should be done to address the uncertainty of the solar influence on the AMOC response.

A cooling caused by the weaker solar activity occurs throughout the middle atmosphere, with a prominent maximum in the mesosphere. Our results indicate an increase in stratospheric $NO_x$ via decreased UV radiation and decrease of mesospheric $NO_x$ as the EPP becomes weaker in a grand solar minimum. Water vapour photolysis is also decreased in the grand solar minimum leading to reduced $HO_x$ concentrations. The declines of $NO_x$ and $HO_x$, together with the reduced UV heating, result in an ozone increase in the mesosphere. In the stratosphere, although the ozone production is reduced here as well due to the decrease in solar UV, the reduction of ODSs cause an increase in ozone.

While this study enhances our understanding of the effect of energetic particle precipitation for high and low energetic particles (such as GCRs and LEEs, respectively), future work should also concentrate on energetic electrons of higher energies (Matthes et al., 2016) and thus evaluate more precisely their effect on future climate. The flux of energetic electrons is dependent on solar activity (e.g. Sinnhuber et al., 2012) and in the grand solar minimum its intensity is diminished. By including these particles in climate models, we can expect an amplification of our results in the grand solar minimum – less $NO_x$ produced and more stratospheric ozone preserved in polar regions, followed by further changes in dynamics and temperature (Arsenovic et al., 2016).

While the future grand solar minimum reduces surface temperature to some degree, it faces us with another problem: thinning of the tropical ozone layer. The acceleration of atmospheric dynamics caused by the warming of tropospheric climate due to the GHGs transports the freshly formed ozone more quickly away from the tropical into extratropical areas and give catalytic chemical cycles less time to deplete ozone. As a consequence, the extratropical areas will reach a "super-recovery" of ozone, while the tropical areas display negative anomalies. Even if the grand solar minimum does not occur, the total ozone in the tropics will be reduced compared to present values. Since the probability of the grand solar minimum to happen in 21$^{st}$ century is rather high (Steinhilber and Beer, 2013), this will compromise the ozone recovery even after a low level of active halogens will be reached. The tropical regions would suffer a loss of up to 6% of the column ozone compared to present values, and tropical ozone would not reach the recovery to the pre-ozone hole (1960-1980) levels. Therefore, all efforts to reduce GHG emissions and the fulfilment of Paris agreement are absolutely crucial. The possibility of failing the Paris climate agreement also brings the risk of thinning of tropical ozone layer.

In the strong and weak solar scenarios, SD and WD, the acceleration of atmospheric dynamics persists throughout the 22$^{nd}$ century, leading to an ozone redistribution from the tropics to the poles, but the global total ozone would stay at the similar levels as at the end of the 21$^{st}$ century. In the SDR and WDR scenarios, when the solar minimum recovers during the 22$^{nd}$ century, global total ozone would increase rapidly and recover (or super-recover).





Stratospheric ozone plays a key role for terrestrial life as it absorbs UV radiation. Although during grand solar minimum UV radiation is decreased, the fact that ozone layer is thinning lets more UV reach the ground. The increase of UV radiation at the ground in grand solar minimum could have implications on terrestrial ecosystem and needs to be investigated in future studies.

Acknowledgements

This work has been supported by the Swiss National Science Foundation under Grant CRSII2-147659 (FUPSOL II) and it is a part of ROSMIC WG1 activity within the SCOSTEP VarSITI program.



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



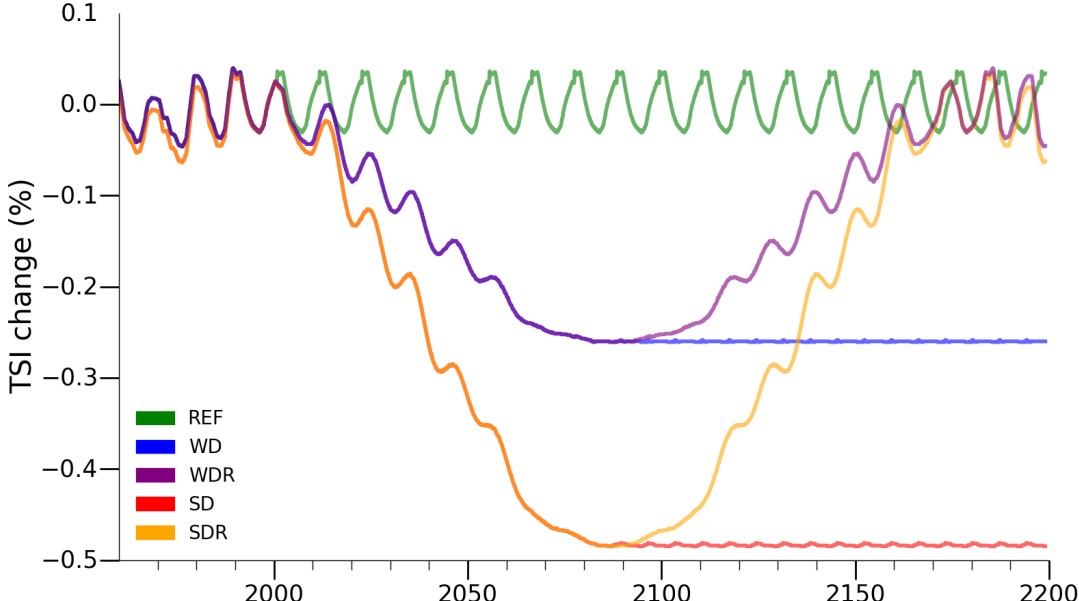

**Figure 1:** Total solar irradiance change relative to the mean of the REF scenario (green line) in % used in the simulations. Weak drop (WD) in blue, weak drop with recovery (WDR) in purple, strong drop (SD) in red and strong drop with recovery (SDR) in orange.





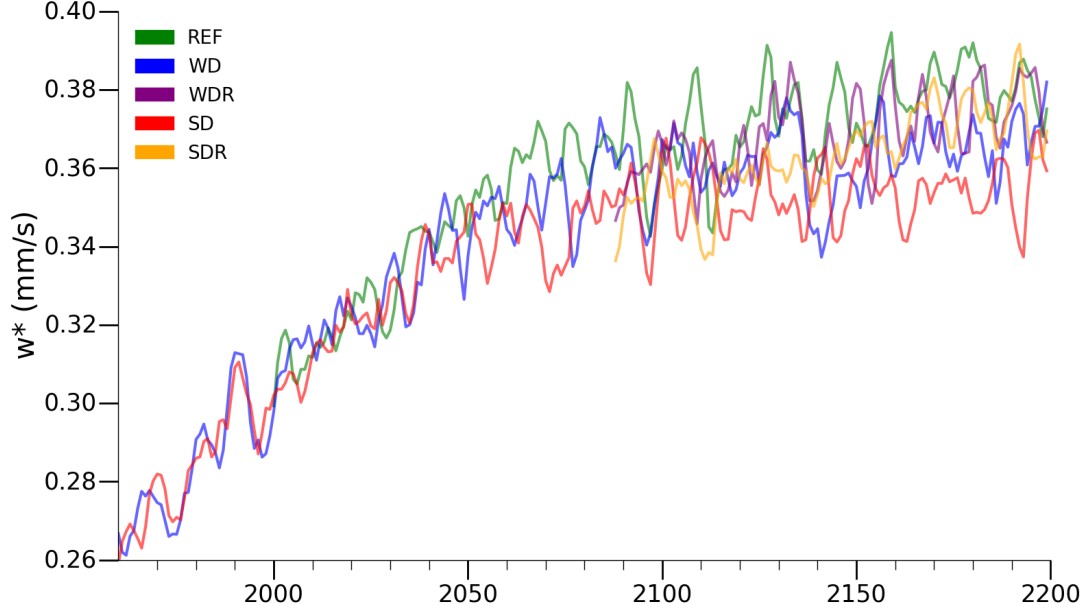

**Figure 2: Annual mean residual vertical velocities (w\*) at 70 hPa averaged over 20° N – 20° S latitudes and smoothed with Savitzky-Golay filter. Weak drop (WD) in blue, weak drop with recovery (WDR) in purple, strong drop (SD) in red and strong drop with recovery (SDR) in orange.**





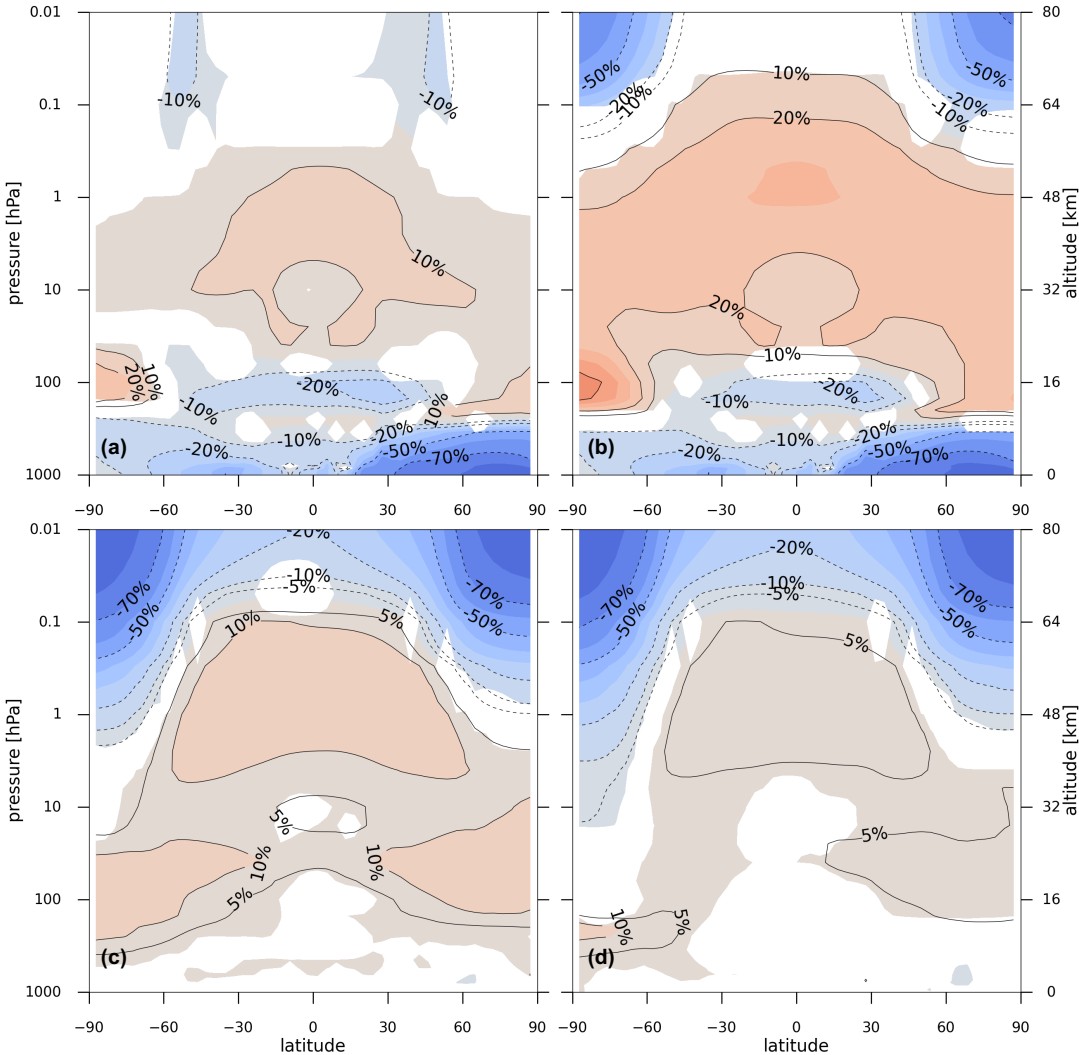

**Figure 3: Annual zonal mean NO$_x$ difference in % of future (2090 – 2099) minus near present (2000 – 2009) for REF (a) and SD (b), and the difference SD – REF (c) and WD – REF (d) under future conditions (2090 – 2099). Coloured regions are significant at the 95% confidence level (calculated using a Student t-test). Colour interval is 10%.**





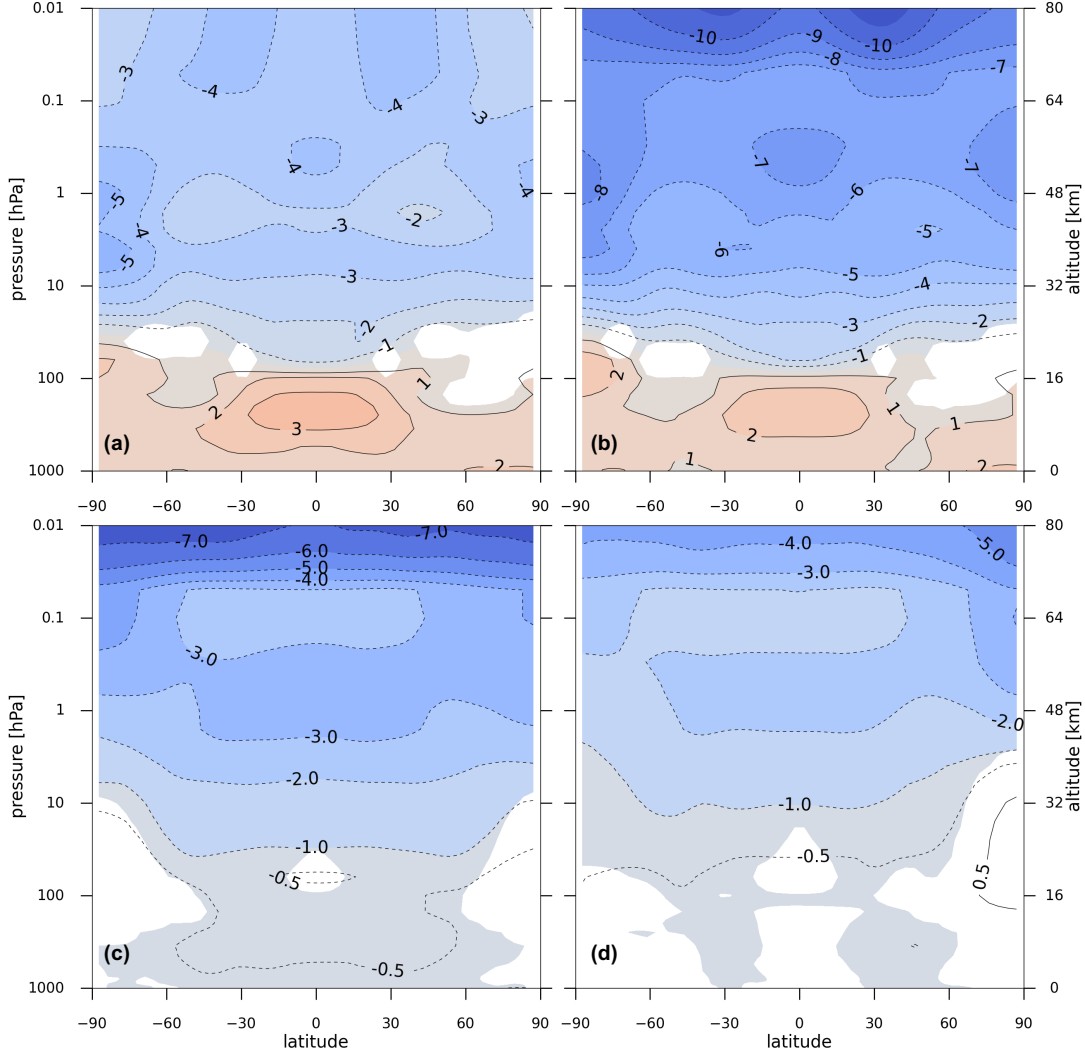

**Figure 4: Annual zonal mean temperature difference in K of future (2090 – 2099) minus near present (2000 – 2009) for REF (a) and SD (b), and difference SD – REF (c) and WD – REF (d) under future conditions (2090 – 2099). Coloured regions are significant at the 95% confidence level (calculated using a Student t-test). Colour interval is 1 K.**



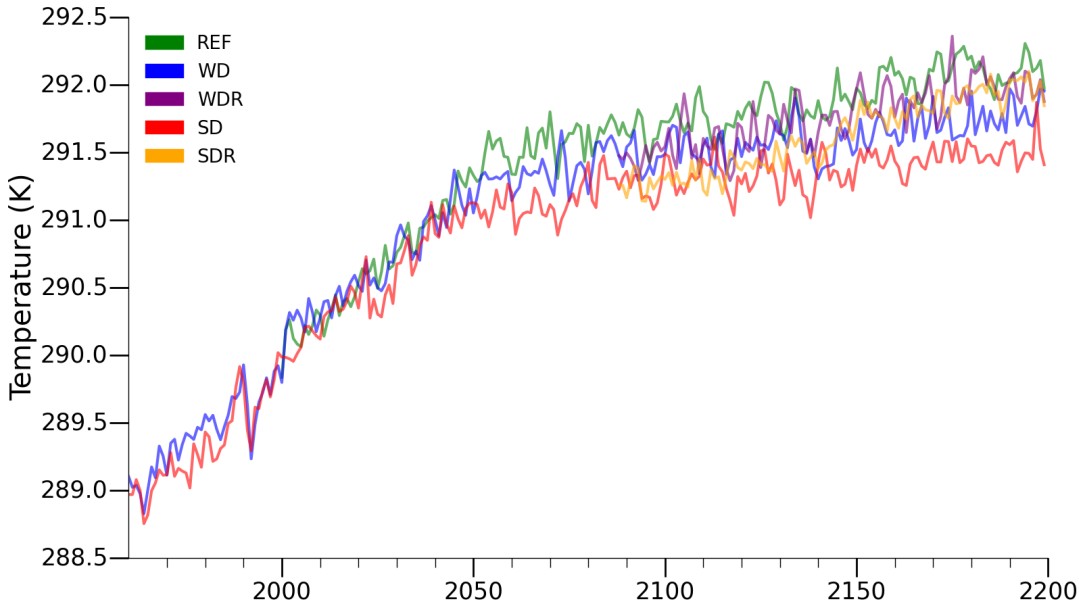

**Figure 5: Annual global mean surface temperature in K of ensemble means from 1960 to 2199.**





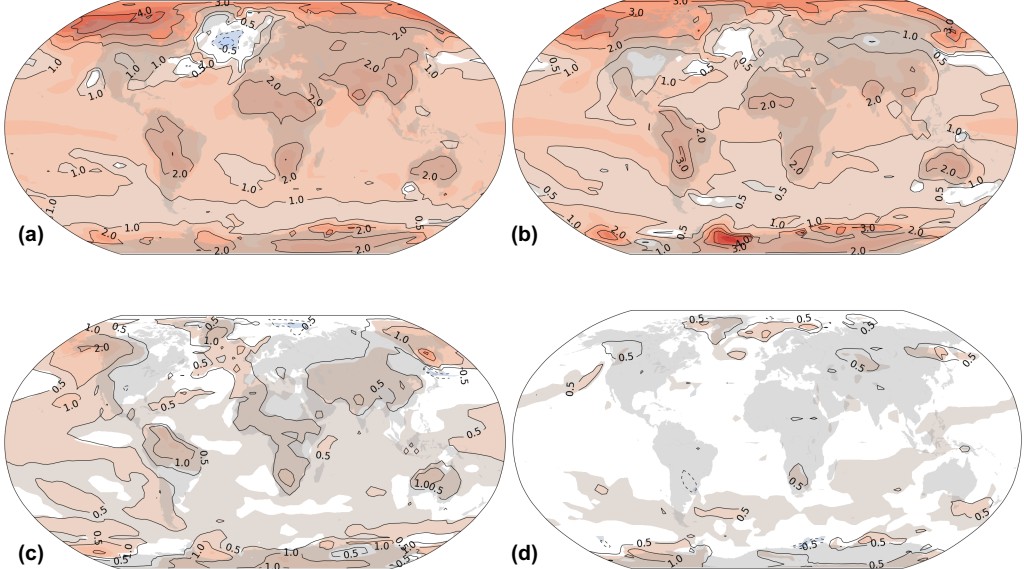

**Figure 6: Spatial distribution of surface temperature difference in K of future (2090 – 2099) minus near present (2000 – 2009) for REF (a) and SD (b), and of far future (2190 – 2199) minus intermediate future (2090 – 2099) for REF (c) and SD (d). Coloured regions are significant at the 95% confidence level (calculated using a Student t-test). Colour interval is 0.5 K.**



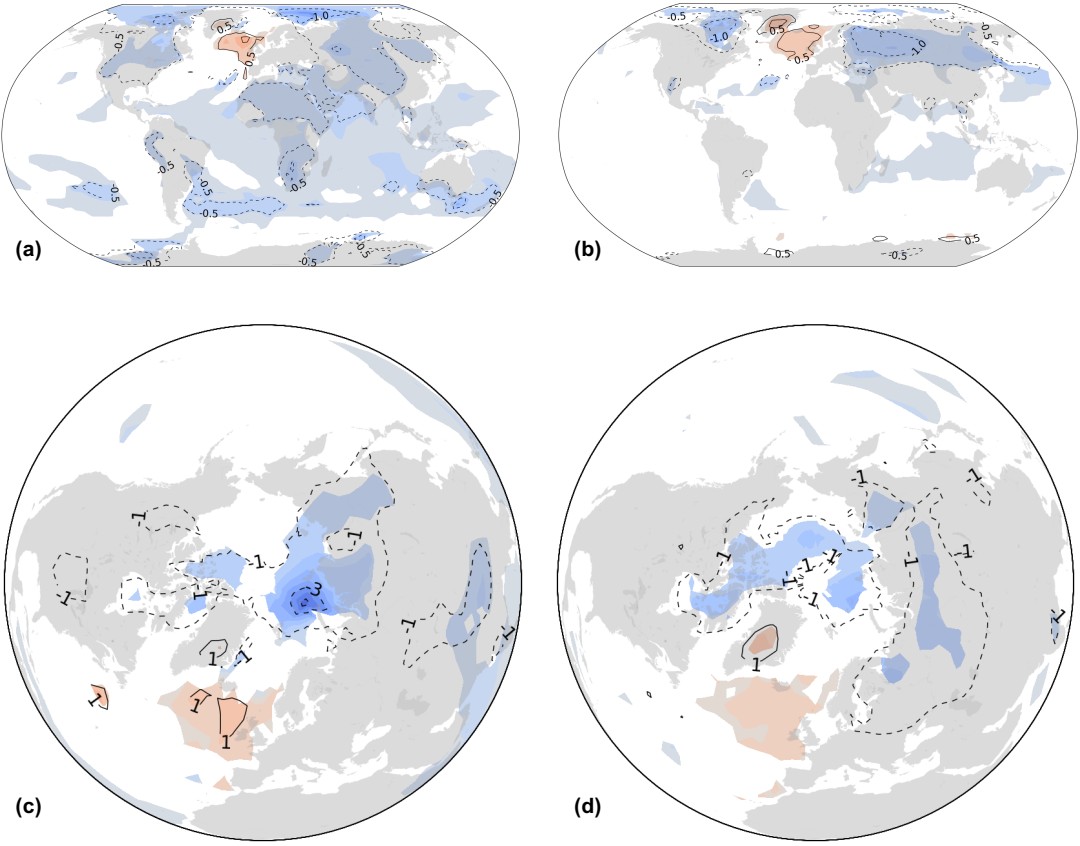

**Figure 7: Global projections of spatial distributions of annual mean surface temperature differences in K of SD minus REF (a) and WD minus REF (b) in the late 21st century (2090 – 2099). Polar projections of boreal winter (DJF) mean surface temperature differences in K of SD minus REF (c) and WD minus REF (d) in the late 21st century (2090 – 2099). Coloured regions are significant at the 95% confidence level (calculated using a Student t-test). Colour intervals are 0.5 K in (a) and (b) and 2 K in (c) and (d).**





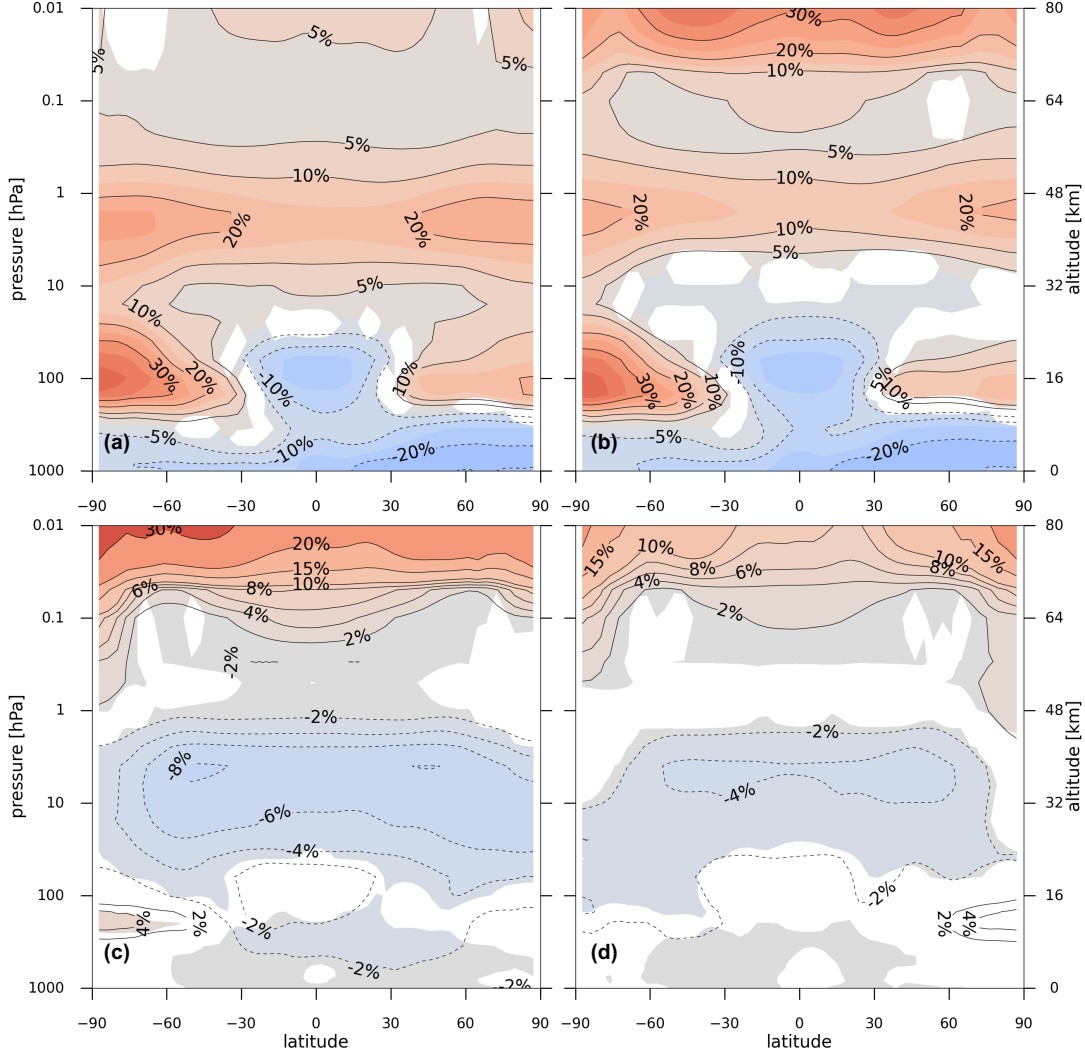

**Figure 8: Annual zonal mean ozone difference of future (2090 – 2009) minus near present (2000 – 2009) for REF (a) and SD (b), and difference of SD minus REF (c) and WD minus REF (d) for the late 21ˢᵗ century (2090 – 2099). Coloured regions are significant at the 95% confidence level (calculated using a Student t-test). Colour interval is 5% in (a) and (b) and 2% in (c) and (d).**

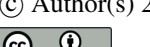


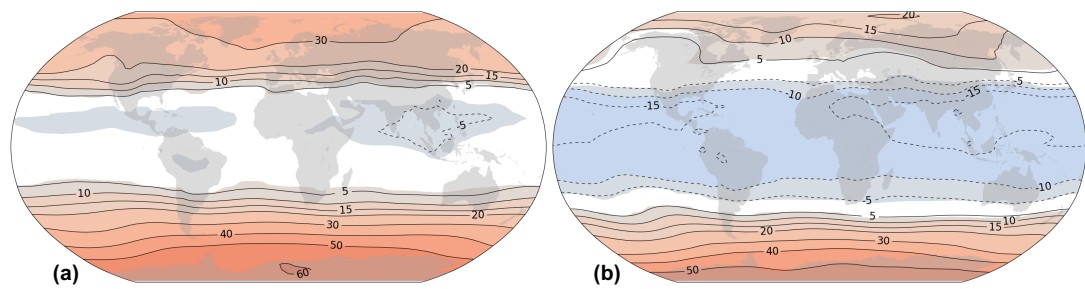

**Figure 9: Spatial distribution of total-column ozone difference in Dobson units of future (2090 – 2099) minus near present (2000 – 2009) conditions for REF (a) and SD (b). Coloured regions are significant at the 95% confidence level (calculated using a Student t-test). Colour interval is 10 Dobson units.**

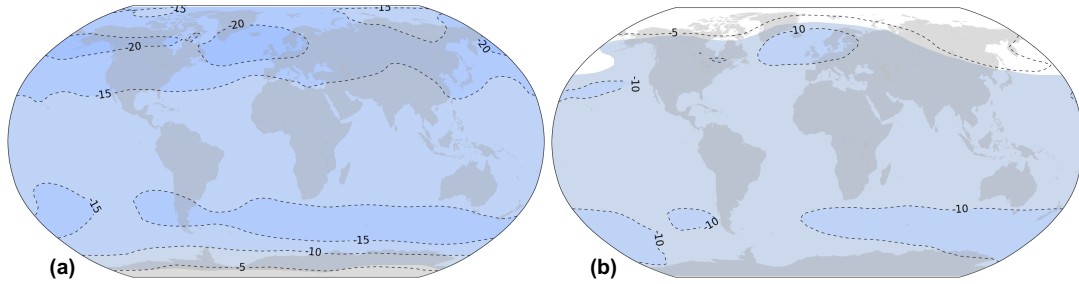

**Figure 10: Spatial distribution of total-column ozone difference in Dobson units of SD minus REF (a) and WD minus REF (b) for the late 21st century (2090 – 2099). Coloured regions are significant at the 95% confidence level (calculated using a Student t-test). Colour interval is 5 Dobson units.**





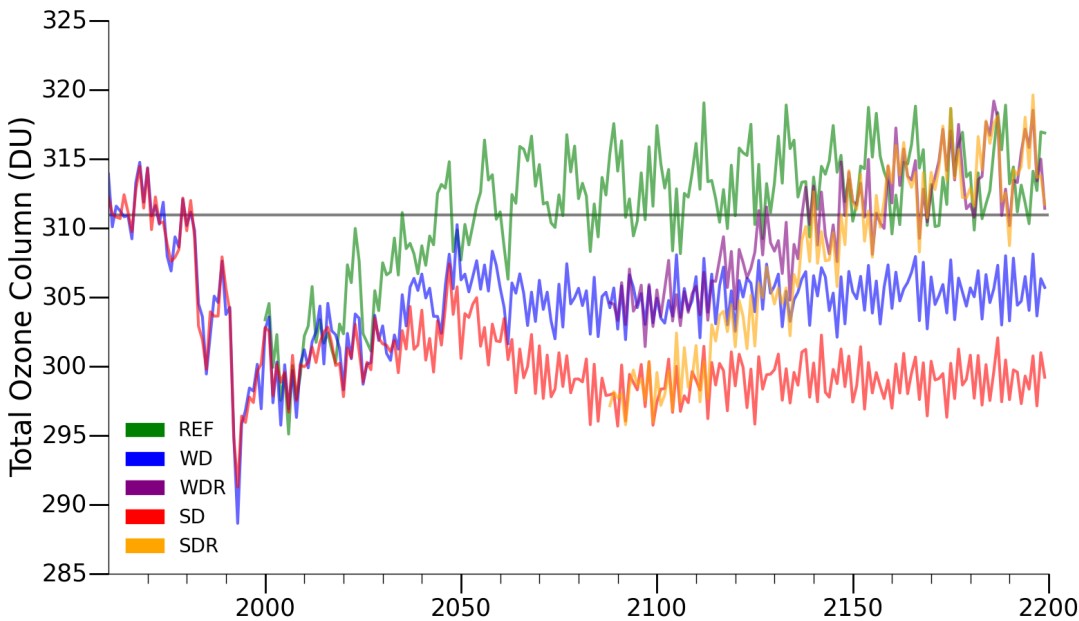

**Figure 11: Annual global mean total-column ozone in Dobson units of ensemble mean values for 1960 – 2199 period. The horizontal grey line presents the 1960 – 1980 period mean value.**