# Peer review of "Implications of potential future grand solar minimum for ozone layer and climate"

_Atmospheric Chemistry and Physics, 2017_

## Referee Comment (RC1) · Anonymous Referee #1 · 5 Nov 2017

**GENERAL COMMENTS**

This is a relatively straightforward paper that makes use of the SOCOL3-MPIOM chemistry-climate model to simulate changes in climate and ozone under strong and weak grand solar minima. The primary conclusion of the paper is that even a strong grand solar minimum, under a very aggressive GHG emissions reduction scenario (RCP2.6) will not completely offset projected increases in global mean surface temperatures.

I have made some minor suggestions for corrections below. Once these have been addressed, the paper will be suitable for publication in Atmospheric Chemistry and Physics.

[Figure]

SPECIFIC COMMENTS

Page 1, line 11: Replace 'greenhouse gas scenario' with 'greenhouse gas emissions scenario'.

Page 1, line 21: I would suggest replacing 'chlorine-induced' with 'halogen-induced' since it is both chlorine and bromine that drives the depletion.

Page 1, line 27: Replace 'of greenhouse' with 'of atmospheric greenhouse'.

Page 2, line 17: I am not sure that this is true. I don't think that the Montreal Protocol prohibits emissions of ODSs into the atmosphere. I think that it only prohibited their production. I encourage the authors to seek clarification on this.

Page 2, line 21: Why the specific focus on terrestrial climate here? Is this not an issue for the ocean also?

Page 3, line 5: The way in which this is written it is not clear whether the sunspots became rare in 1715, or whether they were rare between 1645 and 1715.

Page 3, line 19: I would suggest changing 'cancel global warming' to 'completely offset GHG-induced global warming'.

Page 3, lines 30-31: The statement 'globally cool the surface by around 0.1 K' suggests that the cooling is spatially uniform. Was this indeed the case?

Page 4, line 6: I would suggest 'dissipates' rather than 'recovers' since it is not clear what the grand solar minimum is recovering from.

Figure 2: Why is there a disconnect between the blue and purple lines in 2090 (and similarly for the red and orange lines)? From figure 1 I thought that WD and WDR would be identical up to 2090 and likewise for SD and SDR?

Page 7, lines 13 and 14: I am sure that Zubov (2013) was not the first to show that tropospheric warming is mainly caused by the surface warming due to increase of

down-welling infrared radiation by GHG, enhanced by latent heat release in the middle troposphere. Why did you specifically choose this reference?

Page 7, line 15: I think you need to be clearer what you mean by 'results from increased cooling rates of GHGs' and you should provide a citation to support this assertion.

Page 9, line 14: I think that this would be clearer if you replaced 'show lower surface temperatures' with 'show lower surface temperature reductions'.

Page 9, line 28: If I remember correctly, you had two ensemble members for each simulation. I can't recall that you have said anywhere to this point how they were used. Are the responses shown the average of the ensemble pairs?

Page 10, line 2: You state that the increase in upper stratospheric ozone of 15-20% is a result of reduced intensity of the catalytic ozone destruction by reactive chlorine species. How did you do this attribution solely to chlorine chemistry? Is there no component at all that results from upper stratospheric GHG-induced cooling that slows the $O+O_3 \rightarrow 2O_2$ reaction?

Page 12, line 9: At this point I am wondering how you are treating N2O emissions in your simulations. N2O was supposed to be the biggest depleter of ozone through the 21st century. There seems to be no discussion of this in the context of your simulations. I understand that N2O emissions are the same in all of your simulations, but this statement that the dominant effect on ozone will be solar activity made me wonder about the relative contribution of N2O.

GRAMMAR AND TYPOGRAPHICAL ERRORS

I understand that the authors' first language may not be English. The paper would benefit from a thorough grammar check. I have highlighted just a few of the grammatical errors below.

Page 1, line 25: Replace 'since preindustrial' with 'since the preindustrial'.

Page 1, line 26: Replace 'The mean global' with 'The global mean' or otherwise be specific about which mean you are talking about.

Page 2, line 4: Replace 'RCP8.5' with 'for RCP8.5'.

Page 2, line 6: Replace 'under United' with 'under the United'.

Page 2, line 23: Delete 'basically'.

Page 3, line 30: Replace 'the reduction' with 'a reduction'.

Page 5, line 2: Replace 'assumed previous' with 'assumed in previous'.

Page 6, line 13: Replace 'increase of GHG concentrations' with 'rate of increase in GHG concentrations'.

Page 7, line 16: Replace 'ODSs emissions' with 'ODS emissions'.

Page 8, line 20: Replace 'predict a warming about 2 K' with 'predict a warming of about 2 K'.

Page 9, line 31: Replace 'in polar lower stratosphere' with 'in the polar lower stratosphere'.

Page 10, line 18: Replace 'ODSs concentrations' with 'ODS concentrations'.

Page 12, line 27: Replace 'to conclusion' with 'to conclude'.

---

## Referee Comment (RC2) · Anonymous Referee #2 · 16 Nov 2017

The paper reports on the potential effects of a future solar minimum on climate change projections, according to the SOCOL chemistry-climate model. The sensitivity of future climate (including the evolution of the ozone layer) on the amplitude and duration of the solar minimum are investigated. In agreement with previous studies, it is shown that even a large minimum involving a TSI decrease of up to 6 W/m**2 would only partially (10-20%) ameliorate the warming due to GHG. Moreover, the effects on surface climate are relatively short-lived, as the climate system would quickly bounce back after a "recovery" to present-day solar activity levels. On the other hand, the sensitivity of the evolution of the ozone layer on assumed solar forcing is strikingly large: this is the key and most interesting aspect of this paper. The results presented in the paper

are sufficiently supported by the analysis, and the conclusions are of interest to the wider modeling community. Considering this, I think this paper would be suitable for publication. However, the layout of the paper needs some polishing. Also, some claims are not sufficiently justified. I therefore recommend an extensive set of minor revisions, as detailed below.

MINOR POINTS

Abstract

L9 - "Understanding potential interferences with natural forcings". "interference" sounds strange in this context. I would rephrase this to "Understanding the role of natural forcings in modulating global warming"

L11 remove "several"

L12 "but with different solar forcings" -> a range of different solar forcings

L13 "year 2199, whereas the grand solar..." -> ...year 2199. This reference is compared with grand solar minimum simulations, assuming...

L14 "different durations" -> specify what durations

L14 "decreased ... cooling" -> not clear which of the solar minimums is meant here; please be more specific

L16 "projected to decrease" -> suggest adding this to this sentence: "with respect to a simulation assuming a repeating solar cycle 23"

L16 On the global scale the reduced... -> On the global scale, a reduced...

L16 The regional effects are predicted to be stronger -> would say "significant" instead of "stronger". It is obvious that regional effects are going to be larger than the global counterpart, so i would just say significant to emphasize the importance of regional changes

[Figure]

L19 "In the stratosphere... by up to 8%" it would be good to state the magnitude of the UV forcing (e.g. as W/m2 for the UV spectrum), given the importance of the chemical effect on O3

L22 "completion of ... later". Does this mean that a minimum lasting all the way until 2200 is necessary to get the recovery...? not clear here

L25 remove "current"

L14 p2 "the global ozone depletion" -> the decline in global ozone concentrations

L25-29 I would suggest reducing this discussion... since it is not important for the paper

L33 The effects of the Dalton minimum on Europe are subject to debate, as the minimum is just too short and weak to significantly affect climate. Moreover, there is a problem of interference with volcanoes. Hence, I would rather limit the discussion to the Maunder Minimum (which is already speculative enough), without the need to discussing weaker (and even more speculative) past solar minima.

L3 p3: "computed" -> simulated

L15p3 "is important for a realistic representation of ozone" -> of ozone... variability? The (climatological) ozone layer can be well represented even without the EPP forcing (see CCMVal models that don't include EPP).

L16p3: Suggest removing "With respect to surface temperature".

L18p3: "estimate" -> assess

L19p3: "...would slow or even cancel global warming" -> would lead to possible reduction in the projected warming (I don't think anybody so far has shown that a solar minimum could cancel GW, but only partially ameliorate it)

L22p3: "Meehl..." suggest specifically saying what magnitude and duration for the minimum they chose. There is a bit of spread across different studies in regards to the

amplitude and duration of the imposed "idealized" future minima. So, since there is no unified and accepted definition of "grand solar minimum", these details are needed when reporting the results of a specific study

L24p3: "lower by several tenths..." please always clarify that "lower" is with respect to a reference which is still warming... so we still have a warming even with the minimum, but just smaller in magnitude

L25p3: (in between "grand minimum..." and "However...") I think it would be good to add reference to Chiodo et al., 2016 here, since they used the same global model as Meehl et al, but imposed a more conservative minimum and found important regional effects. Hence, I would add the following... "A follow-up study using the same model but a more conservative solar minimum found important regional effects in the Northern high latitudes, suggesting a reduction of the Arctic Amplification (Chiodo et al., 2016)".

L28p3: "by Meehl et al. (2013)" -> add "Chiodo et al., (2016)" ; they show the same order of magnitude in terms of cooling, at regional level, so they deserve to be cited here, too.

L30p3: "0.13%"... how much is this in W/m**2? Please specifically say it here (e.g. in parenthesis)

L4p4: The novel aspect with respect to Anet et al 2013 is that the minimum is extended in time and the full extent of the response over the 21 and 22 century are analyzed. This should be more clearly stated here... otherwise, there is risk of seeing this paper as a simple "extension of Anet 2013"

L5p4: is there a specific reason as to why this amplitude and duration is chosen? A justification would be nice...

L15p4: "hydrostatic and Bousinnesq" : these two are mutually exclusive; adding a bousinnesq assumption implies the model is not necessarily hydrostatic.

L21p4 applied -> imposed

L22p4 Figure 1: please att W/m2 values to your plot, e.g. as a Y axis on the right hand side

L23p4: "weak drop" : would not call 0.25% a "weak drop", since it is much larger than the 11-yr solar cycle and any current version of TSI reconstructions back to MM... it is weak in the context of the other experiments of this paper though... so I would add "relatively" before "weak" to emphasize this

L3p5 "Ineson et al., 2015..." Chiodo et al., 2016 missing in this load of citations

L3p5: "As described by Meehl et al., 2013" -> Meehl does not really discuss this aspect, so i would drop this citation

L11p5: "In agreement with M2013" -> I would drop this citation. " In agreement" is for results, not for an assumption.... so i would either drop it, or rephrase it to "As in Meehl et al 2013, we emphasize..."

L11p5: "an actual" -> a hypothetical. We are not certain a minimum is actually going to happen to would use "hypothetical" here

L17p5: "solar minimum values." Indicate by which year this recovery is reached

L18p5 "identically to" -> as in

L24p5 "CMIP4" -> there is no CMIP4... either CMIP3 or CMIP5

L4p6: "to elucidate the role of solar forcing" -> to elucidate the role of solar forcing in modulating GHG driven temperature trends

GENERAL COMMENT ON THE RESULTS SECTION: I recommend swapping the order of sections 3.1 and 3.3. I suggest moving 3.3 to 3.1, and moving the BDC section to 3.2, then NOx in 3.3 and ozone in 3.4. Temperature is generally the first variable we look at, and then go back in the causality chain (WSTAR could explain T trends, and then CHEMISTRY)

L9p6: is this filter really needed? What does the t-series look like without such filter?

L14p6: "however... strength, but statistically significant" -> at a reduced rate, although still statistically significant

L16p6 BDC -> the BDC

p19 p6 applied -> imposed

L19p6 " are decreasing and... 21st century" -> are projected to decrease, and N2O will increase during the 21st century.

L29p6: well visible -> clearly visible

L12p7: in the future -> would remove this; REF already implies this is "future projection"

L13p7: "Zubov... middle troposphere". I think this is a generally accepted explanation, based on complex and also simpler (moist GCMs / acquaplanet ) models, so I don't think Zubov is the first one giving this explanation. You could perhaps cite the IPCC instead

L15p7 rates of GHGs -> arising from increased GHG

L15p7 "the secondary maximum" -> Please specify what location you mean here; i.e. the warming at 100 hPa at 70-90S.

L22p7 "has relatively" -> has a relatively

L1p8 "The temperature anomaly... REF" -> suggest rephrasing to: The impact of SD forcing relative to GHG is quantified as the difference between SD and REF: this is shown in Fig.4c

L3p8 "As expected... magnitude". As expected, based on the UV forcing in WD being 50% smaller than SD? If so, please specifically say it here...

L6-L12p8 would put all this at the beginning of the results section

L14p8 "of the minimum" -> from a solar minimum

L15-L19p8 but what is the impact of SDR over the 21st century? Does it make sense to look at the effects 30 years after the recovery...?

L25p8 "reproduces the polar amplificiation well" I would not use the word "reproduce... well" since this is a projection, not a validation against OBS data. Recommend re-wording this to "...simulates a polar amplification too".

L32p8 "see Figure .... SD scenario" -> In this scenario, the model suggests a...

L3p9 "Beyond the scope..., this" rephrase to -> However, this phenomenon needs to be investigated in more detail....

L10p9 "previous century" -> 21st century

L15p9 "see Fig.7a... scenario" -> as shown in Figure 7a for the strong reduction (SD) scenario

L16p9 "but to a smaller magnitude" -> but to a smaller extent

L17p9 "amounting to around" -> and is around

L18p9 "simulating a solar anomaly" -> assuming a future solar minimum

L19p9 "Considering... Asia" -> In the SD scenario, the largest cooling during boreal winter (DJF) is seen over the Barents Sea and northern Asia,

L21p9 Similar cooling areas appear -> Similar cooling appears...

L22p9 "the boreal winter projection" ->boreal winter

L25p9 "Chiodo et al 2016 -> Similarly, Chiodo et al., (2016) reported...

L26p9 "This cooling..." -> In their model, this cooling...

GENERAL COMMENT ON SECTION 3.4 This section on the O3 should follow the NOx analysis (3.2). The sequence of surface temp. BDC, NOx and O3 would also follow

more "naturally", as you start off with the pure thermodynamical response, then look at the dynamics, and then looking into the chemistry.

L7p10: is there a paper showing this point? if there is, please include a citation to it.

L17p10: "low level of solar UV" -> by decreased UV input

L22p10 "Together with the NOx" -> the link to NOx is further evidence that this section would naturally follow immediately after the one on NOx (3.2)

L27-L28p10 -> isn't this a repetition of L20-22 one paragraph above...?

L29p10 suggest adding ", consistent with the smaller (by a factor of 2) UV forcing."

L10p11 "Reduction of... future" -> A future reduction in solar activity

L23-24p11: is this effect statistically significant? The impact of solar irradiance on the polar vortex is generally small and not really significant in models...

L4p12 by several years (Anet et al., 2013).

L5p12: :" we show annual..." -> we show the annual

L8p12 "the dominant ... activity" -> solar activity turns into the the dominant driver of ozone changes.

L18p12 ", but would still be" -> . However, it would still be

L20p12 "While... the year 2100" -> A solar minimum, assuming a very large drop in solar irradiance (SD scenario) is predicted to compensate about 15% of the GHG induced warming by 2100. However, this fraction could increase to about... L32p12 "period" -> season

L9p13 cause -> causes

L11-L13p13 : GENERAL COMMENT: well, the effects of UV vs EPP have not really been separated, as they are lumped together in the forcing imposed in these runs.

Hence, it cannot be really stated that this study "improves our understanding of the effect of EPP"... so either rephrase it, or remove this sentence

L19p13 "it faces us with" -> it poses

L20p13 "The accelleration... dynamics" -> there is no "accelleration of atmospheric dynamics", but of the BDC... so rephrase this to "The accelleration of the BDC"

L3p14 "lets more UV reach the ground" -> allows more UV to reach the ground

---

## Author Comment (AC1) · 10 Jan 2018

We would like to sincerely thank the reviewers. Their comments and suggestions significantly improved the paper quality and readability. We have included/addressed all the comments below and corrected the typos.

Anonymous Referee #1

GENERAL COMMENTS

This is a relatively straightforward paper that makes use of the SOCOL3-MPIOM chemistry-climate model to simulate changes in climate and ozone under strong and weak grand solar minima. The primary conclusion of the paper is that even a strong grand solar minimum, under a very aggressive GHG emissions reduction scenario

(RCP2.6) will not completely offset projected increases in global mean surface temperatures.

I have made some minor suggestions for corrections below. Once these have been addressed, the paper will be suitable for publication in Atmospheric Chemistry and Physics.

SPECIFIC COMMENTS

Page 1, line 11: Replace 'greenhouse gas scenario' with 'greenhouse gas emissions scenario'.

We put "... greenhouse gas concentration scenario RCP4.5" as suggested.

Page 1, line 21: I would suggest replacing 'chlorine-induced' with 'halogen-induced' since it is both chlorine and bromine that drives the depletion.

Fixed. We put "halogen-induced".

Page 1, line 27: Replace 'of greenhouse' with 'of atmospheric greenhouse'.

Fixed. We have done as suggested.

Page 2, line 17: I am not sure that this is true. I don't think that the Montreal Protocol prohibits emissions of ODSs into the atmosphere. I think that it only prohibited their production. I encourage the authors to seek clarification on this.

Yes, the reviewer is right. We changed "emissions" with "production".

Page 2, line 21: Why the specific focus on terrestrial climate here? Is this not an issue for the ocean also?

Good point, "terrestrial and aquatic" would be appropriate, but this phrase usually stands before "ecosystem". We have changed it with "Earth's" instead.

Page 3, line 5: The way in which this is written it is not clear whether the sunspots became rare in 1715, or whether they were rare between 1645 and 1715.

Fixed. Sentence is changed to "A grand solar minimum, which was even more prolonged than the Dalton Minimum was the Maunder Minimum, the period between approximately 1645 and 1715 when sunspots were exceedingly rare."

Page 3, line 19: I would suggest changing 'cancel global warming' to 'completely offset GHG-induced global warming'.

Fixed. We have done as suggested.

Page 3, lines 30-31: The statement 'globally cool the surface by around 0.1 K' suggests that the cooling is spatially uniform. Was this indeed the case?

Good point, now this sentence is "...would decrease global mean temperature by around 0.1 K..."

Page 4, line 6: I would suggest 'dissipates' rather than 'recovers' since it is not clear what the grand solar minimum is recovering from.

Fixed. We have done as suggested.

Figure 2: Why is there a disconnect between the blue and purple lines in 2090 (and similarly for the red and orange lines)? From figure 1 I thought that WD and WDR would be identical up to 2090 and likewise for SD and SDR?

They are identical up to 2090 but the Savitzky-Golay filter makes this disconnect. Please see the plot below that shows SD and SDR w* without the filter.

Page 7, lines 13 and 14: I am sure that Zubov (2013) was not the first to show that tropospheric warming is mainly caused by the surface warming due to increase of down-welling infrared radiation by GHG, enhanced by latent heat release in the middle troposphere. Why did you specifically choose this reference?

Good point, we have changed this reference to IPCC.

Page 7, line 15: I think you need to be clearer what you mean by 'results from increased

cooling rates of GHGs' and you should provide a citation to support this assertion.

We rewrote the sentence to "The temperature decrease in the stratosphere and mesosphere comes from increased cooling rates due to the GHGs rise (IPCC, 2013)".

Page 9, line 14: I think that this would be clearer if you replaced 'show lower surface temperatures' with 'show lower surface temperature reductions'.

Fixed. We have done as suggested.

Page 9, line 28: If I remember correctly, you had two ensemble members for each simulation. I can't recall that you have said anywhere to this point how they were used. Are the responses shown the average of the ensemble pairs?

They are ensemble means (written in the figure captions). We have added this information for all figures as well.

Page 10, line 2: You state that the increase in upper stratospheric ozone of 15-20% is a result of reduced intensity of the catalytic ozone destruction by reactive chlorine species. How did you do this attribution solely to chlorine chemistry? Is there no component at all that results from upper stratospheric GHG-induced cooling that slows the $O+O_3 \rightarrow 2O_2$ reaction?

Good point. The ozone increase is due to the reduced rate of ozone depleting cycles – chlorine, $NO_x$, $HO_x$ and also $O+O_3 \rightarrow 2O_2$ reaction. We have rephrased this statement to: "The increase in the upper stratosphere of 15 – 20% is a result of reduced intensity of the ozone destruction cycles." Further discussion describes this in more details.

Page 12, line 9: At this point I am wondering how you are treating N2O emissions in your simulations. N2O was supposed to be the biggest depleter of ozone through the 21st century. There seems to be no discussion of this in the context of your simulations. I understand that N2O emissions are the same in all of your simulations, but this statement that the dominant effect on ozone will be solar activity made me wonder about the relative contribution of N2O.

N2O itself is not threat to ozone, but it reacts with O(1D) and gives NO which is an ozone depleter. We are discussing NOx in a separate chapter. Although we can't distinguish between NOx coming from tropospheric N2O and energetic particle precipitation, NOx coming from N2O is included in the projections and discussions.

The below errors in the manuscript have been corrected.

GRAMMAR AND TYPOGRAPHICAL ERRORS

I understand that the authors' first language may not be English. The paper would benefit from a thorough grammar check. I have highlighted just a few of the grammatical errors below.

Page 1, line 25: Replace 'since preindustrial' with 'since the preindustrial'.

Page 1, line 26: Replace 'The mean global' with 'The global mean' or otherwise be specific about which mean you are talking about.

Page 2, line 4: Replace 'RCP8.5' with 'for RCP8.5'.

Page 2, line 6: Replace 'under United' with 'under the United'.

Page 2, line 23: Delete 'basically'.

Page 3, line 30: Replace 'the reduction' with 'a reduction'.

Page 5, line 2: Replace 'assumed previous' with 'assumed in previous'.

Page 6, line 13: Replace 'increase of GHG concentrations' with 'rate of increase in GHG concentrations'.

Page 7, line 16: Replace 'ODSs emissions' with 'ODS emissions'.

Page 8, line 20: Replace 'predict a warming about 2 K' with 'predict a warming of about 2 K'.

Page 9, line 31: Replace 'in polar lower stratosphere' with 'in the polar lower stratosphere'.

Page 10, line 18: Replace 'ODSs concentrations' with 'ODS concentrations'.

Page 12, line 27: Replace 'to conclusion' with 'to conclude'.

———————————————————

[Figure]

**Fig. 1.**

---

## Author Comment (AC2) · 10 Jan 2018

We would like to sincerely thank the reviewers. Their comments and suggestions significantly improved the paper quality and readability. We have included/addressed all the comments below and corrected the typos.

Anonymous Referee #2

Review's report on Arsenovic et al.

The paper reports on the potential effects of a future solar minimum on climate change projections, according to the SOCOL chemistry-climate model. The sensitivity of future climate (including the evolution of the ozone layer) on the amplitude and duration of the solar minimum are investigated. In agreement with previous studies, it is shown that

even a large minimum involving a TSI decrease of up to 6 W/m\*\*2 would only partially (10-20%) ameliorate the warming due to GHG. Moreover, the effects on surface climate are relatively short-lived, as the climate system would quickly bounce back after a "recovery" to present-day solar activity levels. On the other hand, the sensitivity of the evolution of the ozone layer on assumed solar forcing is strikingly large: this is the key and most interesting aspect of this paper. The results presented in the paper are sufficiently supported by the analysis, and the conclusions are of interest to the wider modeling community. Considering this, I think this paper would be suitable for publication. However, the layout of the paper needs some polishing. Also, some claims are not sufficiently justified. I therefore recommend an extensive set of minor revisions, as detailed below.

MINOR POINTS

Abstract

L9 - "Understanding potential interferences with natural forcings". "interference" sounds strange in this context. I would rephrase this to "Understanding the role of natural forcings in modulating global warming"

Fixed. We have done as suggested.

L11 remove "several"

Fixed. We have done as suggested.

L12 "but with different solar forcings" -> a range of different solar forcings

Fixed. We have done as suggested.

L13 "year 2199, whereas the grand solar..." -> ...year 2199. This reference is compared with grand solar minimum simulations, assuming...

Fixed. We have done as suggested.

L14 "different durations" -> specify what durations

". . . that last either until 2199 or recover in the 22nd century" is added.

L14 "decreased ... cooling" -> not clear which of the solar minimums is meant here; please be more specific

"by 6.5 W/m2" is inserted in the text.

L16 "projected to decrease" -> suggest adding this to this sentence: "with respect to a simulation assuming a repeating solar cycle 23"

Fixed. "compared to the reference" is added.

L16 On the global scale the reduced... -> On the global scale, a reduced...

Fixed. We have done as suggested.

L16 The regional effects are predicted to be stronger -> would say "significant" instead of "stronger". It is obvious that regional effects are going to be larger than the global counterpart, so i would just say significant to emphasize the importance of regional changes

Fixed. We have done as suggested.

L19 "In the stratosphere... by up to 8%" it would be good to state the magnitude of the UV forcing (e.g. as W/m2 for the UV spectrum), given the importance of the chemical effect on O3

Good point. We have included information of UV reduction in Abstract: ". . .reduction of around 15% of. . ." but also in the Methods: "The drop in the part of the UV spectrum that is most important for ozone production (180 – 250 nm) is about 9% in WD and WDR and about 15% in SD and SDR."

L22 "completion of ... later". Does this mean that a minimum lasting all the way until 2200 is necessary to get the recovery...? not clear here

No, it will recover as soon as the minimum recovers. We have removed "in the 22nd century or later". L25 remove "current"

Fixed. We have done as suggested.

L14 p2 "the global ozone depletion" -> the decline in global ozone concentrations

Fixed. We have done as suggested.

L25-29 I would suggest reducing this discussion... since it is not important for the paper

We have reduced the discussion to sentence: "Apart from 11-year solar cycle, the solar activity oscillates in the cycles in the order of hundred years, called "grand solar minima" and "grand solar maxima"."

L33 The effects of the Dalton minimum on Europe are subject to debate, as the minimum is just too short and weak to significantly affect climate. Moreover, there is a problem of interference with volcanoes. Hence, I would rather limit the discussion to the Maunder Minimum (which is already speculative enough), without the need to discussing weaker (and even more speculative) past solar minima.

It is true that Dalton minimum interferes with volcanoes, but it has been shown that it affected global mean temperature (Anet et al., 2013, ACP). This paragraph serves as motivation to say that the grand solar minima are able to impact global mean temperature.

L3 p3: "computed" -> simulated

Fixed. We have done as suggested.

L15p3 "is important for a realistic representation of ozone" -> of ozone... variability? The (climatological) ozone layer can be well represented even without the EPP forcing (see CCMVal models that don't include EPP).

Fixed. We have done as suggested.

L16p3: Suggest removing "With respect to surface temperature".

Fixed. We have done as suggested.

L18p3: "estimate" -> assess

Fixed. We have done as suggested.

L19p3: "...would slow or even cancel global warming" -> would lead to possible reduction in the projected warming (I don't think anybody so far has shown that a solar minimum could cancel GW, but only partially ameliorate it)

Fixed. We have done as suggested.

L22p3: "Meehl..." suggest specifically saying what magnitude and duration for the minimum they chose. There is a bit of spread across different studies in regards to the amplitude and duration of the imposed "idealized" future minima. So, since there is no unified and accepted definition of "grand solar minimum", these details are needed when reporting the results of a specific study

We have included this information in the introduction section: "with total solar irradiance drop of about 3.9 W/m2 (0.25%) in 2024 – 2065 period".

L24p3: "lower by several tenths..." please always clarify that "lower" is with respect to a reference which is still warming... so we still have a warming even with the minimum, but just smaller in magnitude

Fixed. We put "than the reference" to be more specific.

L25p3: (in between "grand minimum..." and "However...") I think it would be good to add reference to Chiodo et al., 2016 here, since they used the same global model as Meehl et al, but imposed a more conservative minimum and found important regional effects. Hence, I would add the following... "A follow-up study using the same model but a more conservative solar minimum found important regional effects in the Northern high latitudes, suggesting a reduction of the Arctic Amplification (Chiodo et al., 2016)".

Thank you very much for the suggestion. We have added sentence in the manuscript "A follow-up study using the same model but a more conservative solar minimum found important regional effects in the Northern high latitudes, suggesting a reduction of the Arctic Amplification (Chiodo et al., 2016)".

L28p3: "by Meehl et al. (2013)" -> add "Chiodo et al., (2016)" ; they show the same order of magnitude in terms of cooling, at regional level, so they deserve to be cited here, too.

Added. Thank you.

L30p3: "0.13%"... how much is this in W/m\*\*2? Please specifically say it here (e.g. in parenthesis)

We added "... about 1.75 W/m2 (0.13%)..." to be more specific.

L4p4: The novel aspect with respect to Anet et al 2013 is that the minimum is extended in time and the full extent of the response over the 21 and 22 century are analyzed. This should be more clearly stated here... otherwise, there is risk of seeing this paper as a simple "extension of Anet 2013"

Thank you for the suggestion. We have included "The novel aspect with respect to Anet et al 2013 is that the minimum is extended in time and the full extent of the response over the 21st and 22nd century are analysed."

Sentence "Here we investigate the atmospheric response to a potential grand solar minimum which starts around 2020, reaches full depth by about 2090, and lasts either until 2200 or recovers within the 22nd century." is removed because the minimum details are discussed later.

L5p4: is there a specific reason as to why this amplitude and duration is chosen? A justification would be nice...

We have removed that sentence (please see the previous comment)

L15p4: "hydrostatic and Bousinnesq": these two are mutually exclusive; adding a bousinnesq assumption implies the model is not necessarily hydrostatic.

We removed "hydrostatic and Bousinnesq" and connected this sentence with the following: "The oceanic component is MPIOM, a primitive equation model which includes a dynamic/thermodynamic sea-ice module and uses a curvilinear orthogonal grid which allows for various setups."

L21p4 applied -> imposed

Fixed. We have done as suggested.

L22p4 Figure 1: please att W/m2 values to your plot, e.g. as a Y axis on the right hand side

We have added the change in TSI in W/m2 as left axis and percentage axis as right.

L23p4: "weak drop" : would not call 0.25% a "weak drop", since it is much larger than the 11-yr solar cycle and any current version of TSI reconstructions back to MM... it is weak in the context of the other experiments of this paper though... so I would add "relatively" before "weak" to emphasize this

Fixed. We have added "relatively".

L3p5 "Ineson et al., 2015..." Chiodo et al., 2016 missing in this load of citations

We have added Chiodo reference as suggested.

L3p5: "As described by Meehl et al., 2013" -> Meehl does not really discuss this aspect, so i would drop this citation

We have removed Meehl reference and added Schrijver et al (2011) and Foukal et al (2011). The sentence now is: "Previous estimates regarding the TSI decrease during the Maunder Minimum compared to present-day values range from somewhere close to present 11-year solar minima (Schrijver et al., 2011), to reductions of 0.15% to 0.3%

below present solar minima (Foukal et al., 2011) all the way to more than 0.4% below present solar minima derived by Shapiro et al. (2011) and applied here in the SD and SDR scenarios."

L11p5: "In agreement with M2013" -> I would drop this citation. " In agreement" is for results, not for an assumption.... so i would either drop it, or rephrase it to "As in Meehl et al 2013, we emphasize..."

Fixed. The sentence construction is changed to "As in. . ."

L11p5: "an actual" -> a hypothetical. We are not certain a minimum is actually going to happen to would use "hypothetical" here

Fixed. We have done as suggested.

L17p5: "solar minimum values." Indicate by which year this recovery is reached

Fixed. We have added ". . .about 2170 – 2180"

L18p5 "identically to" -> as in

Fixed. We have done as suggested.

L24p5 "CMIP4" -> there is no CMIP4... either CMIP3 or CMIP5

Fixed. We changed CMIP4 to Community Climate System Model 3 (CCSM3).

L4p6: "to elucidate the role of solar forcing" -> to elucidate the role of solar forcing in modulating GHG driven temperature trends

Fixed. We have included ". . . in modulating GHG driven temperature trends"

GENERAL COMMENT ON THE RESULTS SECTION: I recommend swapping the order of sections 3.1 and 3.3. I suggest moving 3.3 to 3.1, and moving the BDC section to 3.2, then NOx in 3.3 and ozone in 3.4. Temperature is generally the first variable we look at, and then go back in the causality chain (WSTAR could explain T trends, and then CHEMISTRY)

We have changed the order of the chapters as suggested by reviewer.

L9p6: is this filter really needed? What does the t-series look like without such filter?

Please see the plot below (yearly means). The filter makes the differences in w* in scenarios more obvious, especially when there are 5 scenarios shown.

L14p6: "however... strength, but statistically significant" -> at a reduced rate, although still statistically significant

Fixed. We have done as suggested.

L16p6 BDC -> the BDC

Fixed. We have done as suggested.

p19 p6 applied -> imposed

Fixed. We have done as suggested.

L19p6 " are decreasing and... 21st century" -> are projected to decrease, and N2O will increase during the 21st century.

Fixed. We have changed this sentence to "... surface emissions of NOx are projected to decrease and concentrations of N2O will increase during the 21st century..."

L29p6: well visible -> clearly visible

Fixed. We have done as suggested.

L12p7: in the future -> would remove this; REF already implies this is "future projection"

Fixed. We have done as suggested.

L13p7: "Zubov... middle troposphere". I think this is a generally accepted explanation, based on complex and also simpler (moist GCMs / acquaplanet ) models, so I don't think Zubov is the first one giving this explanation. You could perhaps cite the IPCC instead

We have changed Zubov et al reference to IPCC, 2013

L15p7 rates of GHGs -> arising from increased GHG

We changed this sentence to "The temperature decrease in the stratosphere and mesosphere comes from increased cooling rates due to the GHGs rise"

L15p7 "the secondary maximum" -> Please specify what location you mean here; i.e. the warming at 100 hPa at 70-90S.

We included "... around 100 hPa". It is already said that it is around Antarctica.

L22p7 "has relatively" -> has a relatively

Fixed. We have done as suggested.

L1p8 "The temperature anomaly... REF" -> suggest rephrasing to: The impact of SD forcing relative to GHG is quantified as the difference between SD and REF: this is shown in Fig.4c

We changed the manuscript as suggested.

L3p8 "As expected... magnitude". As expected, based on the UV forcing in WD being 50% smaller than SD? If so, please specifically say it here...

We put "... due to the lower UV forcing...". It is not exactly 50% - it is 9% in WD and 15% in SD.

L6-L12p8 would put all this at the beginning of the results section

Thank you for the suggestion. We followed the advice, this paragraph is in the beginning of the chapter.

L14p8 "of the minimum" -> from a solar minimum

Fixed. We have done as suggested.

L15-L19p8 but what is the impact of SDR over the 21st century? Does it make sense
to look at the effects 30 years after the recovery...?

SD and SDR are very similar in the 21st century (identical up until 2087). Therefore, we don't discuss SDR in the 21st century. We look into the effects after the recovery to see if the minimum would slow down warming and ozone layer recovery before the sun goes back to present values.

L25p8 "reproduces the polar amplificiation well" I would not use the word "reproduce... well" since this is a projection, not a validation against OBS data. Recommend rewording this to "...simulates a polar amplification too".

We have changed the manuscript accordingly.

The below points are all included as suggested:

L32p8 "see Figure .... SD scenario" -> In this scenario, the model suggests a...

L3p9 "Beyond the scope..., this" rephrase to -> However, this phenomenon needs to be investigated in more detail....

L10p9 "previous century" -> 21st century

L15p9 "see Fig.7a... scenario" -> as shown in Figure 7a for the strong reduction (SD) scenario

L16p9 "but to a smaller magnitude" -> but to a smaller extent

L17p9 "amounting to around" -> and is around

L18p9 "simulating a solar anomaly" -> assuming a future solar minimum

L19p9 "Considering... Asia" -> In the SD scenario, the largest cooling during boreal winter (DJF) is seen over the Barents Sea and northern Asia,

L21p9 Similar cooling areas appear -> Similar cooling appears...

L22p9 "the boreal winter projection" ->boreal winter

L25p9 "Chiodo et al 2016 -> Similarly, Chiodo et al., (2016) reported...

L26p9 "This cooling..." -> In their model, this cooling...

GENERAL COMMENT ON SECTION 3.4 This section on the O3 should follow the NOx analysis (3.2). The sequence of surface temp. BDC, NOx and O3 would also follow more "naturally", as you start off with the pure thermodynamical response, then look at the dynamics, and then looking into the chemistry.

Good idea, we agree and change the order as suggested.

L7p10: is there a paper showing this point? if there is, please include a citation to it.

These lines are: "Conversely, the future decline of NOx surface emissions will result in less tropospheric ozone with a maximum in the northern hemisphere of up to 20%." We are not aware of any other study showing this claim.

L17p10: "low level of solar UV" -> by decreased UV input

Fixed. We have done as suggested.

L22p10 "Together with the NOx" -> the link to NOx is further evidence that this section would naturally follow immediately after the one on NOx (3.2)

We have changed the order of paragraphs as suggested.

L27-L28p10 -> isn't this a repetition of L20-22 one paragraph above...?

It is not a repetition: L20-22 refer to future decrease of HOx and lines L27-28 to decrease of HOx in case of solar minimum. Both result in ozone increase.

L29p10 suggest adding ", consistent with the smaller (by a factor of 2) UV forcing."

We have changed the manuscript as suggested.

L10p11 "Reduction of... future" -> A future reduction in solar activity

Fixed. We have done as suggested.

L23-24p11: is this effect statistically significant? The impact of solar irradiance on the polar vortex is generally small and not really significant in models...

We found the significant zonal wind change (please see the plot below).

Figure. Annual zonal mean zonal wind difference in m/s of SD minus REF under future conditions (2090 – 2099)

The below lines are all included as suggested by reviewer:

L4p12 by several years (Anet et al., 2013).

L5p12: :" we show annual..." -> we show the annual

L8p12 "the dominant ... activity" -> solar activity turns into the the dominant driver of ozone changes.

L18p12 ", but would still be" -> . However, it would still be

L20p12 "While... the year 2100" -> A solar minimum, assuming a very large drop in solar irradiance (SD scenario) is predicted to compensate about 15% of the GHG induced warming by 2100. However, this fraction could increase to about...

L32p12 "period" -> season

L9p13 cause -> causes

L11-L13p13 : GENERAL COMMENT: well, the effects of UV vs EPP have not really been separated, as they are lumped together in the forcing imposed in these runs. Hence, it cannot be really stated that this study "improves our understanding of the effect of EPP"... so either rephrase it, or remove this sentence

We have rephrased the sentence to: ". . . While this study includes the effect of energetic particle precipitation. . ."

The below lines are all included as suggested by reviewer:

L19p13 "it faces us with" -> it poses

L20p13 "The accelleration... dynamics" -> there is no "accelleration of atmospheric dynamics", but of the BDC... so rephrase this to "The accelleration of the BDC"

L3p14 "lets more UV reach the ground" -> allows more UV to reach the ground

[Figure]

**Fig. 1.**

[Figure]

**Fig. 2.**